# Combining mutation and recombination statistics to infer clonal families in antibody repertoires

Natanael Spisak[1], Gabriel Athènes[1,2], Thomas Dupic[3], Thierry Mora[1]*, Aleksandra M Walczak[1]*

[1]Laboratoire de physique de l'École normale supérieure, CNRS, PSL University, Sorbonne Université and Université de Paris, Paris, France; [2]Saber Bio SAS, Institut du Cerveau, iPEPS The Healthtech Hub, Paris, France; [3]Department of Organismic and Evolutionary Biology, Harvard University, Cambridge, United States

**Abstract** B-cell repertoires are characterized by a diverse set of receptors of distinct specificities generated through two processes of somatic diversification: V(D)J recombination and somatic hypermutations. B-cell clonal families stem from the same V(D)J recombination event, but differ in their hypermutations. Clonal families identification is key to understanding B-cell repertoire function, evolution, and dynamics. We present HILARy (high-precision inference of lineages in antibody repertoires), an efficient, fast, and precise method to identify clonal families from single- or paired-chain repertoire sequencing datasets. HILARy combines probabilistic models that capture the receptor generation and selection statistics with adapted clustering methods to achieve consistently high inference accuracy. It automatically leverages the phylogenetic signal of shared mutations in difficult repertoire subsets. Exploiting the high sensitivity of the method, we find the statistics of evolutionary properties such as the site frequency spectrum and $d_N/d_S$ ratio do not depend on the junction length. We also identify a broad range of selection pressures spanning two orders of magnitude.

*For correspondence:
thierry.mora@gmail.com (TM);
aleksandra.walczak@phys.ens.
fr (AMW)

## Editor's evaluation

This fundamental study provides a new, high-performance algorithm for B-cell clonal family inference. The new algorithm is highly innovative and based on a rigorous probabilistic analysis of the relevant biological processes and their imprint on the resulting sequences. The strength of evidence regarding the algorithm's performance is convincing, as the algorithm has been benchmarked against two state-of-the-art methods for clonal family inference on two synthetic data sets generated with two independent, state-of-the-art methods for B cell repertoire simulation. This work will be fundamental to immunologists and important to any researcher or clinician utilizing B cell receptor repertoires in their field.

## Introduction

B cells play a key role in the adaptive immune response through their diverse repertoire of immunoglobulins (Ig). These proteins recognize foreign pathogens in their membrane-bound form (called B-cell receptor [BCR]), and battle them in their soluble form (antibody). Each B cell expresses a unique BCR that can bind their antigenic targets with high affinity. The set of distinct BCR harbored by the organism is highly diverse (*Briney et al., 2019*), thanks to two processes of diversification: V(D)J recombination and somatic hypermutation. These stochastic processes ensure that repertoires can

**Figure 1.** Clonal families and **VJ**$l$ classes. (**A**) Variable region of the immunoglobulin heavy chain (IgH)-coding gene. (**B**) A clonal family is a lineage of related B cells stemming from the same VDJ recombination event. The partition of the B-cell receptor (BCR) repertoire into clonal families is a refinement of the partition into **VJ**$l$ classes, defined by sequences with the same V and J usage and the same complementarity determining region 3 (CDR3) length $l$. (**C–D**) Properties of **VJ**$l$ classes in donor 326651 from **Briney et al., 2019**. (**C**) Distribution of **VJ**$l$ class sizes exhibits power-law scaling. The total number of pairwise comparisons in the largest **VJ**$l$ classes is $\sim 10^{5^2} = 10^{10}$. (**D**) Distribution of the CDR3 length $l$. The distribution is in yellow for in-frame CDR3 sequences ($l$ multiple of 3), and in gray for out-of-frame sequences.

match a variety of potential threats, including proteins of bacterial and viral origin that have never been encountered before.

V(D)J recombination takes place during B-cell differentiation (**Hozumi and Tonegawa, 1976**; **Schatz and Swanson, 2011**). For each Ig chain, V, D, and J gene segments for the heavy chain, and V and J gene segments for the light chain, are randomly chosen and joined with random non-templated deletions and insertions at the junction, creating a long, hypervariable region, called the complementarity determining region 3 (CDR3) (**Figure 1A**). Cells are subsequently selected for the binding properties of their receptors and against autoreactivity. At this stage, the repertoire already covers a wide range of specificities. In response to antigenic stimuli, B cells with the relevant specificities are recruited to germinal centers, where they proliferate and their Ig-coding genes undergo somatic hypermutation (**Victora and Nussenzweig, 2022**) in the process of affinity maturation. Somatic hypermutation consists primarily of point substitutions, as well as insertions and deletions, restricted to Ig-coding genes (**Feng et al., 2020**). The mutants are selected for high affinity to the particular antigenic target, and the best binders further differentiate into plasma cells and produce high-affinity antibodies. A more diverse pool of variants forms the memory repertoire, leaving an imprint of the immune response that can be recalled upon repeated stimulation.

A clonal family is defined as a collection of cells that stem from a unique V(D)J rearrangement, and has diversified as a result of hypermutation, forming a lineage (**Figure 1B**). These families are the main building blocks of the repertoire. Since members of the same family usually share their specificities (**De Boer et al., 2001**), affinity maturation first competes families against each other for antigen binding in the early stages of the reaction, and then selects out the best binders within families in the later stages (**Tas et al., 2016**; **Mesin et al., 2016**).

High-throughput sequencing of single receptor chains offers unprecedented insight into the diversity and dynamics of the repertoire. Recent experiments have sampled the repertoires of the immunoglobulin heavy chain (IgH) of healthy individuals at great depth to reveal their structure (*Briney et al., 2019*). Disease-specific cohorts are now routinely subject to repertoire sequencing studies, which help to quantify and understand the dynamics of the B-cell response (*Kreer et al., 2020*; *Nielsen et al., 2020*).

Partitioning BCR repertoire sequence datasets into clonal families is a critical step in understanding the architecture of each sample and interpreting the results. Identifying these lineages allows for quantifying selection (*Yaari and Uduman, 2012*; *Yaari and Kleinstein, 2015*; *Ruiz Ortega et al., 2023*) and for detecting changes in longitudinal measurements (*Nielsen et al., 2020*; *Turner et al., 2020*). In recent years, many strategies have been developed that take advantage of CDR3 hypervariability (*Abdollahi et al., 2020*): it is generally unlikely that the same or a similar CDR3 sequence be generated independently multiple times (*Elhanati et al., 2015*; *Ruiz Ortega et al., 2023*). Other approaches make use of the information encoded in the intra-lineage patterns of divergence due to mutations (*Briney et al., 2016*; *Nouri and Kleinstein, 2020*). All inference techniques need to balance accuracy and speed. Simpler methods are fast but have low precision (also called positive predictive value) while more complex algorithms have long computation times that do not scale well with the number of sequences. This prohibits the analysis of recent large-scale data such as *Briney et al., 2019*.

In this work, we propose a new method for inferring clonal families from high-throughput sequencing data that is both fast and accurate. We use probabilistic models of junctional diversity to estimate the level of clonality in repertoire subsets, allowing us to tune the sensitivity threshold a priori to achieve a desired accuracy. We have developed two complementary algorithms. The first one (HILARy-CDR3) uses a very fast CDR3-based approach that avoids pairwise comparisons, while the second one (HILARy-full) additionally exploits information encoded in the phylogenetic signal outside of the junction. We compare our method with state-of-the-art approaches in a benchmark with realistic synthetic data.

**Table 1.** Summary of notations used throughout the paper.

Hats ˆ denote estimates from the fit of the mixture model. Stars ∗ denote estimates after imposing 99% precision. The 'post' subscript denotes quantities after applying single-linkage clustering to obtain a partition from positive pairs.

|  | Definition |
|---|---|
| $\rho$ | Prevalence/fraction of positive pairs |
| $\pi$ | Precision = TP/(TP+FP) |
| $s$ | Sensitivity = TP/(TP+FN) |
| $p$ | Fallout = FP/(FP+TN) |
| $t$ | Threshold on CDR3 distance |
| $l$ | CDR3 length |
| $n$ | CDR3 Hamming distance of a pair |
| $x$ | Normalized CDR3 Hamming distance= $l/n$ |
| $x'$ | CDR3 Hamming distance, centered and scaled |
| $y'$ | Shared mutations on V segment, centered and scaled |
| $\mu$ | Mean $x$ between positive pairs |
| $P_{\mathrm{T}}$ | Model distribution for positive pairs |
| $P_{\mathrm{F}}$ | Model distribution for negative pairs |

## Results

### Analysis of pairwise distances within VJ$l$ classes

A common strategy for partitioning a BCR repertoire dataset into clonal families is to go through all pairs of sequences and identify pairs of clonally related sequences. In the following, we call such related pairs *positive*, and pairs of sequences belonging to different families *negative*. Then, the partition is built by single-linkage clustering, which consists of recursively grouping all positive pairs. Two characteristics of the repertoire complicate the search for this partition: large total number of pairs and low proportion of positive pairs. In this section we analyze and model the statistics of pairs of sequences in natural repertoires to inform our choice of the clustering method and parameters. In the next section we will leverage that analysis to design an optimized clustering procedure. To help following notations, a summary of their definitions is provided in *Table 1*.

A pair of related sequences is expected to share the same V and J genes, as well as the same CDR3 length $l$, as determined by alignment to the templates (*Figure 1A*). The methods developed here begin by partitioning the data into VJ$l$ classes, defined as subsets of sequences with the same V and J gene usage, and CDR3 length $l$ (*Figure 1B*). For a description of the data preprocessing and alignment to the V and J gene templates, see Methods. Clustering will then be performed within each VJ$l$ class independently. While this first step severely limits the number of unnecessary comparisons, some VJ$l$ classes still exceed $10^5$ sequences in large datasets, leading to the order of $10^{10}$ pairs (see *Figure 1C* for the distribution of the VJ$l$ class sizes $N$ for donor 326651 of *Briney et al., 2019*).

The CDR3 plays an important role in encoding the signature of the VDJ rearrangement. As we will see, the CDR3 length $l$ has a strong impact on the difficulty of clonal family reconstruction. The distribution of CDR3 lengths $l$ observed in the data is shown in *Figure 1D*. In what follows we restrict our analysis to sequences with CDR3 lengths a multiple of 3 and between 15 and 105, relying on the common approximation that sequences with no frameshift in the CDR3 come from a productive naive ancestor. The number of sequences with length larger than 105 is too small to reach meaningful conclusions, and sequences of length smaller than 15 are likely nonfunctional (as evidenced by the similar number of in-frame and out-of-frame sequences in *Figure 1D*).

In each VJ$l$ class, we call *prevalence* and denote by $\rho$ the proportion of positive pairs, i.e., the number of positive pairs divided by the total number of pairs. This quantity is unknown in the absence of the ground-truth partition. However, we can estimate it from the statistics of pairwise distances. We compute the Hamming distance $n$ of each pair of CDR3s, defined as the number of positions at which the two nucleotide sequences differ. The distribution of these distances normalized by the CDR3 length, denoted by $x$, shows a clear bimodal structure in data (donor 326651 of *Briney et al., 2019*), with two identifiable components (*Figure 2A*): the contribution of positive pairs (of proportion $\rho$) peaks near $x = 0$ and decays quickly, whereas the bell-shaped contribution of negative pairs (of proportion $1 - \rho$) peaks around $x = 1/2$.

The prevalence $\rho$ can be formally written as $[\Sigma_i z_i(z_i - 1)/2]/[N(N - 1)/2]$, where $z_i$ denote the sizes of the clonal families in the VJ$l$ class, but we do not know these sizes before the partition into families is found. To overcome this issue, we developed a method to estimate $\rho$ a priori, without knowing the family structure (Methods). We do this by fitting the empirical distribution of $x$ as a mixture model, $P(x) = \hat{\rho}P_{\mathrm{T}}(x) + (1 - \hat{\rho})P_{\mathrm{F}}(x)$, where $P_{\mathrm{T}}(x)$ and $P_{\mathrm{F}}(x)$ are the distributions of distances between positive (T as true) and negative (F as false) pairs (*Figure 2C and D*), estimated as follows. $P_{\mathrm{F}}(x) = P_{\mathrm{F}}(x|l)$ is computed for each length $l$ by generating a large number of unrelated, same-length sequences with the soNNia model of recombination and selection (*Isacchini et al., 2021*), and calculating the distribution of their pairwise distances (Methods). $P_{\mathrm{T}}(x)$ is approximated by a Poisson distribution, $P_{\mathrm{T}}(x) = (\mu l)^{xl}e^{-\mu l}/(xl)!$, with adjustable parameter $\mu$, which is proportional to the average hypermutation rate within the clone. The fit of $P(x)$ by the mixture model is performed for each VJ$l$ class with an expectation-maximization algorithm which finds maximum likelihood estimates of the prevalence $\hat{\rho}$ and mean intra-family distance $\hat{\mu}$, the only free parameters of the mixture model.

The results of the fit to real data (donor 326651 of *Briney et al., 2019*) show that $\hat{\mu}$ varies little between VJ$l$ classes, around $\hat{\mu} \simeq 4\%$ (*Figure 2—figure supplement 1*). In contrast, the prevalence $\hat{\rho}$ varies widely across classes, spanning three orders of magnitude (*Figure 2B*). In addition, when we examine the VJ$l$ classes with increasing CDR3 length $l$, we find that the part of the model distribution corresponding to positive pairs, $P_{\mathrm{T}}(x)$, varies little, whereas the model distribution over negative pairs

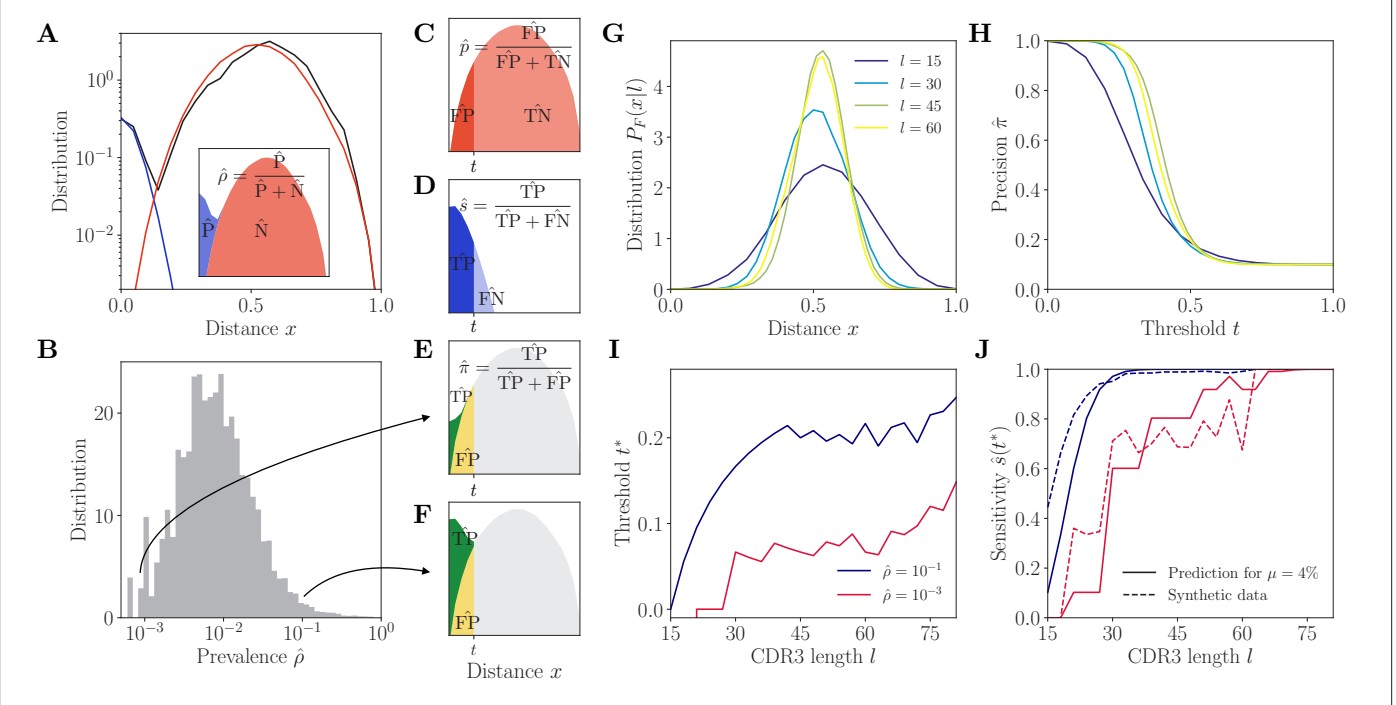

**Figure 2.** Complementarity determining region 3 (CDR3)-based inference method (HILARy-CDR3). (**A**) Example distribution of normalized Hamming distances, $x = n/l$, for one **VJl** class with CDR3 length $l = 21$, V gene IGHV3-9 and J gene IGHJ4 (black). We fit the distribution by a mixture of positive pairs (belonging to the same family, in blue) and negative pairs (belonging to different families, in red). See *Figure 2—figure supplement 5* for example fit results across different CDR3 lengths. Inset: the prevalence is defined as a fraction of positive pairs and was estimated to $\hat{\rho} = 3.1\%$. Data from donor 326651 of *Briney et al., 2019*. (**B**) Distribution of the maximum likelihood estimates of prevalence $\hat{\rho}$ across **VJl** classes in donor 326651. (**C–F**) The choice of threshold $t$ on the normalized Hamming distance $x$ translates to the following a priori characteristics of inference (illustrated here for arbitrarily chosen $\rho$ and $\mu$). (**C**) Fallout rate $\hat{p}(t) = \hat{FP}/(\hat{FP} + \hat{TN})$. The null distribution of all negatives (N=FP + TN) is estimated using the soNNia sequence generation software. (**D**) Sensitivity $\hat{s}(t) = \hat{TP}/(\hat{TP} + \hat{FN})$. (**E–F**) Precision $\hat{\pi} = \hat{TP}/(\hat{TP} + \hat{FP})$. For the same choice of threshold $t$, a low prevalence of $\hat{\rho} = 10^{-3}$ (**E**) leads to lower precision than high prevalence of $\hat{\rho} = 10^{-1}$ (**F**). (**G**) Model distribution $P_T(x|\mu)$ of distances between unrelated sequences, for $l = 15, 30, 45, 60$, computed by the soNNia software. (**H**) Precision $\hat{\pi}$, computed a priori (i.e. before doing the inference) from the model with $\hat{\mu} = 0.04$, $\hat{\rho} = 0.1$, and $l = 15, ..., 81$ (colors as in G), as a function of the threshold $t$. For each **VJl** class and its own inferred $\hat{\rho}$ and $\hat{\mu}$, the threshold $t$ is chosen to achieve a desired $\pi^*$. (**I**) High-precision threshold $t^*$ ensuring $\hat{\pi}(t^*) = \pi^* = 99\%$ a priori, as a function of CDR3 length $l$ for different values of the prevalence $\hat{\rho}$, and $\hat{\mu} = 0.04$, as predicted by the model. (**J**) Sensitivity $\hat{s}(t^*)$ at the high-precision threshold $t^*$, as a function of CDR3 length $l$ for different values of the prevalence $\hat{\rho}$ (colors as in I). Solid lines denote a priori prediction for intermediate mean distance $\mu = 4\%$, dashed lines denote actual performance of HILARy-CDR3 in a synthetic dataset.

The online version of this article includes the following figure supplement(s) for figure 2:

**Figure supplement 1.** Mean intra-family distances.

**Figure supplement 2.** Null distribution $P_N(x|l)$ of CDR3 distances between unrelated sequences for $l \in [15, 81]$, computed by soNNia software.

**Figure supplement 3.** Distribution of normalized Hamming distances.

**Figure supplement 4.** Distribution of post-selection probabilities.

**Figure supplement 5.** Site frequency spectra estimated for families identifed using high-precision CDR3-based inference method (HILARy-CDR3) in the subset of the data where this approach is highly reliable (large-$l$ and large-$\hat{\rho}$ regime).

**Figure supplement 6.** Distribution of normalized Hamming distances $x = n/l$, for $l$ classes, averaging over all **VJl** classes.

**Figure supplement 7.** Prevalence and **VJl** class size.

$P_F(x)$ becomes more and more peaked around 1/2 (*Figure 2* and *Figure 2—figure supplement 2*), making the two categories more easily separable.

## CDR3-based inference method with adaptive threshold

We want to build a classifier between positive and negative pairs using the normalized distance $x$ alone, by setting a threshold $t$ so that pairs are called positive if $x \le t$, and negative otherwise. Using our model for $P(x)$, for any given $t$ we can evaluate the number of true positives ($\hat{TP}$) and false

negatives ($\hat{\text{FN}}$) among all positive pairs ($\hat{\text{P}} = \hat{\text{TP}} + \hat{\text{FN}}$), as well as true negatives ($\hat{\text{TN}}$) and false positives ($\hat{\text{FP}}$) among the negative pairs ($\hat{\text{N}} = \hat{\text{TN}} + \hat{\text{FP}}$), as schematized in *Figure 2—figure supplement 2C and D*.

Our goal is to set a threshold $t$ that ensures a high precision, $\hat{\pi}(t)$, defined as a proportion of true positives among all pairs classified as positive (*Figure 2E*). In a single-linkage clustering approach, we will join two clusters with at least one pair of positive sequences between them. Therefore, it is critical to limit the number of false positives, which can cause the erroneous merger of large clusters. We can write:

$$\hat{\pi}(t) \equiv \frac{\hat{\text{TP}}}{\hat{\text{TP}} + \hat{\text{FP}}} = \frac{\hat{\rho}\hat{s}(t)}{\hat{\rho}\hat{s}(t) + (1 - \hat{\rho})\hat{p}(t)}, \tag{1}$$

$$\hat{p}(t) \equiv \frac{\hat{\text{FP}}}{\hat{\text{N}}} = \sum_{x \leq t} P_{\text{F}}(x), \tag{2}$$

and $\hat{s}(t)$ is the estimated sensitivity (*Figure 2D*), evaluated from the Poisson fit to $P_{\text{T}}$ (Methods):

$$\hat{s}(t) \equiv \frac{\hat{\text{TP}}}{\hat{\text{P}}} = \sum_{x \leq t} P_{\text{T}}(x) \tag{3}$$

Finally, the estimated prevalence $\hat{\rho} \equiv \hat{\text{P}}/(\hat{\text{P}} + \hat{\text{N}})$ is inferred from the $P(x)$ distribution as explained above.

*Figure 2H* shows $\hat{\pi}(t)$ as a function of $t$ for different CDR3 lengths and a fixed value of $\hat{\rho}$. For each VJ$l$ class, we define the threshold $t = t^*$ that reaches 99% precision, $\hat{\pi}(t^*) = \pi^* = 99\%$, by inverting *Equation 1*. This adaptive threshold depends on the *VJl* class through the CDR3 length $l$ and the prevalence $\rho$, and it increases with both (*Figure 2I*): low clonality (small $\rho$) means few positive pairs and a smaller adaptive threshold, while short CDR3 means less information and a stricter inclusion criterion.

The predicted sensitivity, $\hat{s}(t^*)$, which tells us how much of the positives we are capturing, is shown in *Figure 2J*. We conclude that for a wide range of parameters, the method is predicted to achieve both high precision and high sensitivity. However, it is expected to fail when the prevalence and the CDR3 length are both low. At the extreme, for small values $\rho$ and $l$, even joining together identical CDR3s ($t = 0$) results in poor precision because of convergent recombination (reflected by $t^* < 0$).

The resulting procedure, which we call HILARy-CDR3, can be applied to Ig repertoire data through the following steps: (1) group sequences by VJ$l$ class; (2) in each class, fit the mixture model to the distribution of pairwise distance to infer $\hat{\rho}$ and $\hat{\mu}$; (3) invert *Equations 1–3* to find the high-precision threshold $t^*$; (4) classify positive and negative pairs according to that threshold; (5) complete the partition by applying single-linkage clustering to positive pairs.

## Tests on synthetic datasets

So far we have presented a method to set a high-precision threshold with predictable sensitivity, based on estimates from the distribution of distances $P(x)$ only. To verify that these performance predictions hold in a realistic inference task, we designed a method to generate realistic synthetic datasets where the clonal family structure is known. This generative method will also be used in the next sections to create a benchmark for comparing different clustering algorithms.

We first estimated the distribution of clonal family sizes from the data of *Briney et al., 2019*, by applying HILARy-CDR3 with adaptive threshold as described above to VJ$l$ classes for which the inference was highly reliable, i.e. for which the predicted sensitivity was $z \in [10, 100]$. In that limit, clusters are clearly separated and the partition should depend only weakly on the choice of clustering method. The resulting distribution of clone sizes follows a power-law with exponent –2.3.

To create a synthetic lineage, we first draw a random progenitor using the soNNia model for IgH generation (*Figure 2—figure supplement 4*). We then draw the size of the lineage at random, using the power-law distribution above. Mutations are then randomly drawn on each sequence of the lineage in a way that preserves the mutation sharing patterns observed in families of comparable size from the partitioned data (*Figure 2—figure supplement 5*). We thus generated $10^4$ lineages and $2.5 \cdot 10^4$ sequences. Note that, while that procedure is partially based on real data, in particular the distribution of lineage sizes and mutational co-occurence structure in the lineages, it uses completely

random sequences and mutations. In addition, these empirical observables were inferred from $\mathrm{VJ}l$ classes that were easy to cluster, ensuring that they are not biased by our inference method, and therefore should not give it an unfair advantage. More details about the procedure are given in the Methods.

We applied the HILARy-CDR3 method to this synthetic dataset. The sensitivity achieved at $t^*$ roughly follows and sometimes even outperforms the predicted one $\hat{s}(t^*)$ across different values of $\rho$ and $l$ (**Figure 2J**, dashed line), validating the approach and the choice of the adaptive high-precision threshold $t^*$ (the discrepancy is due to the fact that $\mu$ is assumed to be constant in the prediction, while it varies in the dataset). These results also confirm the poor performance of the method at low prevalences and short CDR3s.

## Incorporating phylogenetic signal

To improve the performance of HILARy-CDR3, we set out to include the phylogenetic signal encoded in the mutation spectrum of the templated regions of the sequences. Two sequences belonging to the same lineage are expected to share some part of the mutational histories, and therefore sequences with shared mutations are more likely to be in the same lineage.

We focus on the template-aligned region of the sequence outside of the CDR3, where we can reliably identify substitutions with respect to the germline. We denote the length of this alignment

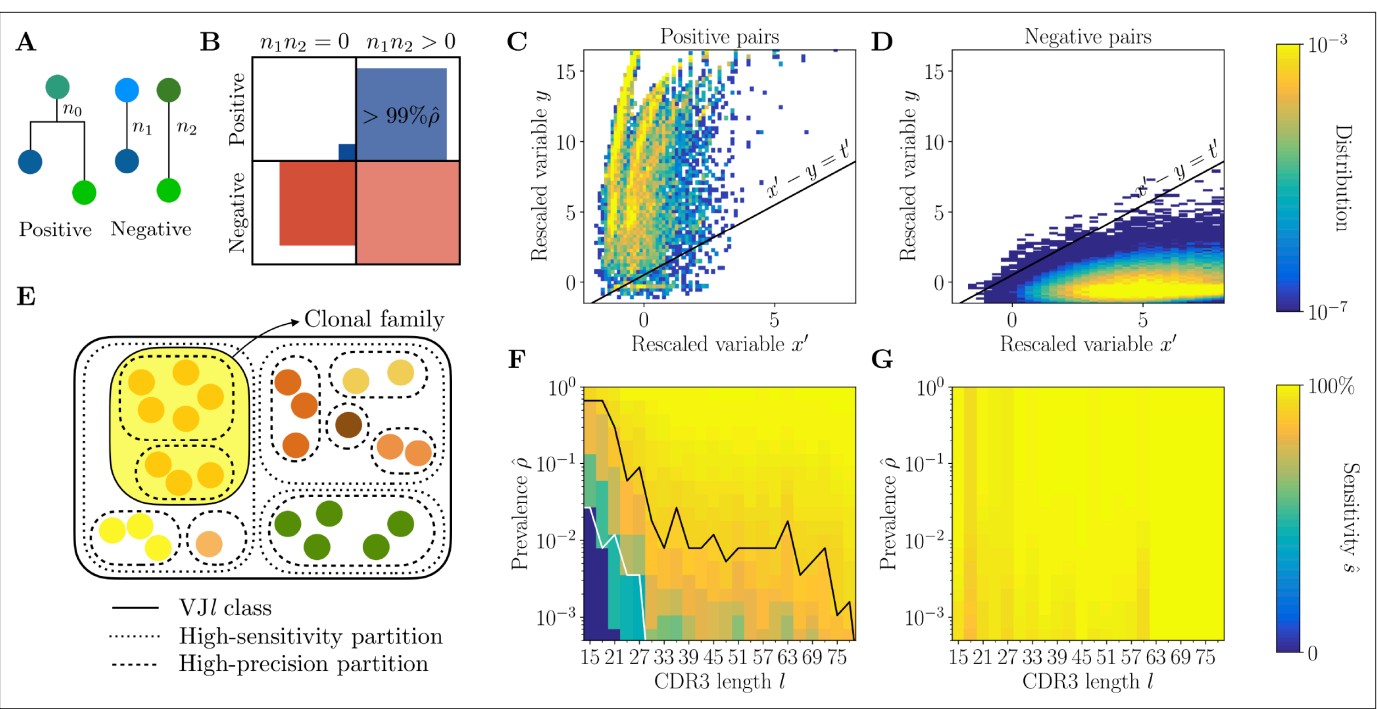

**Figure 3.** Full inference method with mutational information. (**A**) For a pair of sequences, $n_1, n_2$ denote the numbers of mutations along the templated region (V and J), and $n_0$ is the number of shared mutations. For related sequences, $n_0$ corresponds to mutations on the initial branch of the tree, and is expected to be larger than for unrelated sequences, where $n_0$ corresponds to coincidental mutations. (**B**) Positive and negative pairs are called mutated if both sequences have mutations $n_1, n_2 > 0$. Among positive pairs in the synthetic datasets, more than 99% are mutated. (**C, D**) Distributions of the rescaled variables $x'$ and $y$ (**Equation 4**), for pairs of synthetic sequences belonging to the same lineage (positive pairs) and sequences belonging to different lineages (negative pairs). The separatrix $x' - y = t'$ marks a high-precision (99%) threshold choice. (**E**) To limit the number of pairwise comparisons we make use of high-precision and high-sensitivity complementarity determining region 3 (CDR3)-based partitions. High precision corresponds to the choice $t = t^*$. High sensitivity corresponds to a coarser partition where $t$ is set to achieve 90% sensitivity. When the two partitions disagree, mutational information can be used to break the coarse, high-sensitivity partition into smaller clonal families. (**F, G**) Mutations-based methods achieve high sensitivity across all CDR3 lengths $l$ in the synthetic dataset (**G**), extending the range of applicability with respect to the CDR3-based method (**F**).

The online version of this article includes the following figure supplement(s) for figure 3:

**Figure supplement 1.** Merging partitions.

**Figure supplement 2.** Error vs $\mathrm{VJ}l$ class size.

by $L$, so that the total length of the sequence is $l + L$. For each pair of sequences, we define $n_1, n_2$ as the number of mutations along the templated alignment in the two sequences, $n_0$ the number of mutations shared by the two, and $n_L = n_1 + n_2 - 2n_0$ the number of non-shared mutations. Under the hypothesis of shared ancestry, the $n_0$ shared mutations fall on the shared part of the phylogeny, and are expected to be more numerous than under the null hypothesis of independent sequences, where they are a result of random co-occurrence (*Figure 3A*).

To balance the tradeoff between the information encoded in the templated part of the sequence and the recombination junction, we can compute characteristic scales for the two variables of interest: the number of shared mutations and the CDR3 distance $n$. Intuitively, in highly mutated sequences, we can expect substantial divergence in the CDR3. At the same time, the number of mutations in the templated regions would increase, possibly leading to more shared mutations. Conversely, sequences with few or no mutations carry no information in the templated region, but we also expect their CDR3 sequences to be nearly identical. To adapt a clustering threshold to the two variables, we compute their expectations under the two assumptions, and define the rescaled variables

$$x' = \frac{n - \langle n \rangle_{\mathrm{T}}}{\sigma_{\mathrm{T}}(n)}, \quad y = \frac{n_0 - \langle n_0 \rangle_{\mathrm{F}}}{\sigma_{\mathrm{F}}(n_0)}, \tag{4}$$

where $\langle n \rangle_{\mathrm{T}} = l(n_L + 1)/L$ is the expected value of $n$ under the hypothesis that sequences belong to the same lineage (see Methods), and $\langle n_0 \rangle_{\mathrm{F}} = n_1 n_2/L$ is the expected value of $n_0$ under the hypothesis that they do not. The standard deviations are likewise defined as $\sigma_{\mathrm{T}}(n) = \sqrt{\langle n^2 \rangle_{\mathrm{T}} - \langle n \rangle_{\mathrm{T}}^2} = (1/L)\sqrt{l(l + L)(n_L + 1)}$ and $\sigma_{\mathrm{F}}(n_0) = \sqrt{\langle n_0^2 \rangle_{\mathrm{F}} - \langle n_0 \rangle_{\mathrm{F}}^2} = \sqrt{n_1 n_2/L}$ (Methods).

For more than 99% of positive pairs, both sequences are mutated, i.e., $n_1, n_2 > 0$ (*Figure 3B*). Without loss of sensitivity, we focus on the mutated part of the dataset, since we cannot use $y$ for non-mutated sequences. The distributions of $x'$ and $y$ for positive and negative pairs (*Figure 3C and D*) are well separated, with positive pairs characterized by an overrepresentation of shared mutations. By adding the phylogenetic signal $y$ we can identify positive pairs of sequences that have significantly diverged in their CDR3 ($x' > 0$) but share significantly more mutations than expected (large $y$).

Computing $y$ for each pair of sequences is computationally expensive. To avoid examining all pairs, we first perform two different nested clusterings of each VJ$l$ class using the CDR3-based method: the previously described HILARy-CDR3 'fine' partition with threshold $t^*$ that ensures high precision $\hat{\pi} = 99\%$; and a 'coarser' clustering with a high threshold $t = t_{\mathrm{sens}}$ that ensures high estimated sensitivity $\hat{s} = 90\%$ (Methods and *Figure 3E*). When lineages are easily separable (e.g. for sufficiently large prevalence $\rho$ and CDR3 length $l$), these two partitions coincide, and we do not need to compute $y$ at all. When they do not coincide, we can use the phylogenetic signal $y$ to refine the coarse high-sensitivity partition. We only need to compute $y$ for pairs that belong to the same coarse cluster, but not to the same fine cluster: the phylogenetic signal $y$ is used to merge the fine-partition clusters into clonal families (Methods and *Figure 3—figure supplement 1*). This allows us to considerably reduce the number of pairwise comparisons that we need to make between the templated regions of the sequences.

Using $x'$ and $y$, we classify pairs of sequences as positive (i.e. belonging to the same family) if $y \geq x' - t'$, and as negative otherwise. We can compute the expected sensitivity on the synthetic data, and find that it reaches values $\geq 90\%$ across the whole range of prevalence $\rho$ and CDR3 lengths $l$, outperforming HILARy-CDR3 in the low-$\rho$, low-$l$ region (*Figure 3F and G*). This proves that using the phylogenetic signal significantly improves performance over HILARy-CDR3.

The procedure outlined above, which we call HILARy-full, may be summarized as follows: (1) group sequences by VJ$l$ class; (2) apply HILARy-CDR3 twice, once with the high-precision threshold as before to get a fine partition, and once with a high-sensitivity threshold to get a coarse partition, thus obtaining two nested partitions; (3) compute $x'$ and $y$ using *Equation 4* only for pairs that belong to the same coarse cluster but to different fine clusters; (4) merge all fine clusters with at least one pair with $y \geq x' - t'$.

## Benchmark of the methods on heavy-chain datasets

We compare our approach to state-of-the-art methods. In addition to our two algorithms—HILARy-CDR3 and HILARy-full—our benchmark includes the alignment-free method of *Lindenbaum et al., 2021*, partis (*Ralph and Matsen, 2016*), and the spectral clustering method of SCOPer (*Nouri and*

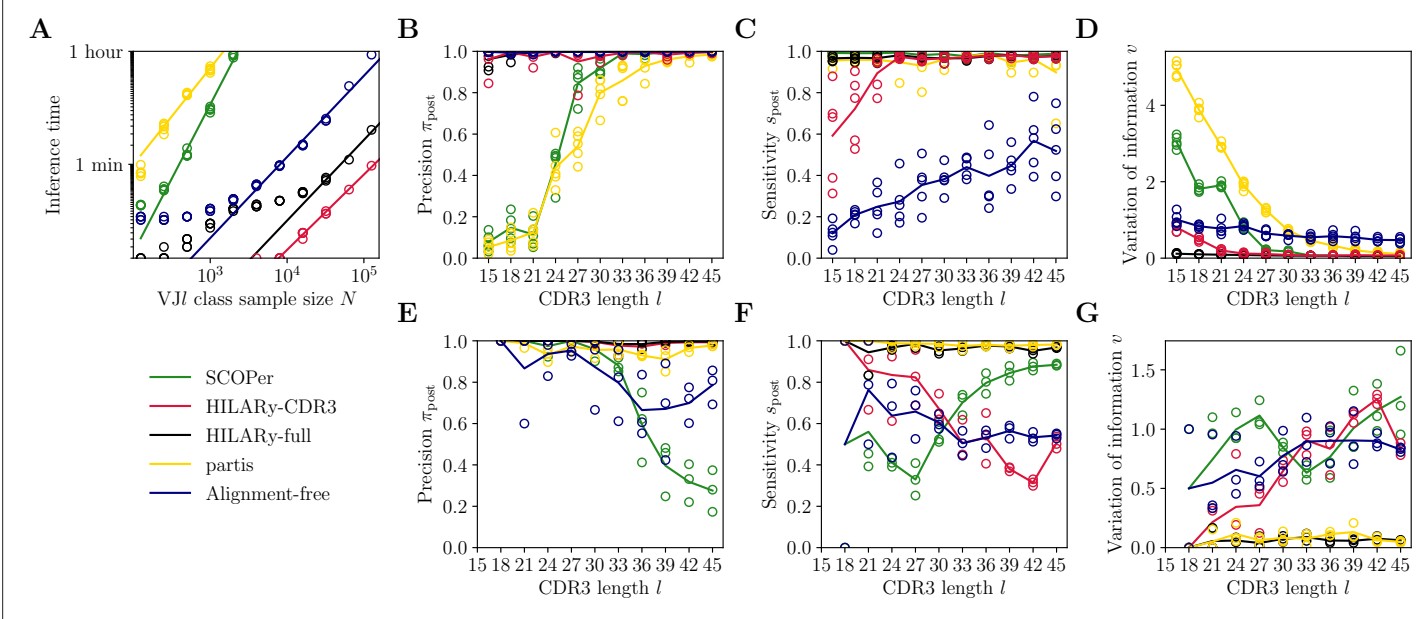

**Figure 4.** Benchmark of the alternative methods on synthetic heavy-chain repertoires. (**A**) Comparison of inference time using subsamples from the largest **VJ**$l$ class found in donor 326651 from **Briney et al., 2019**. Comparisons were done on a computer with 14 double-threaded 2.60 GHz CPUs (28 threads in total) and 62.7 Gb of RAM. (**B**) Clustering precision $\pi_{post}$ (post single-linkage clustering of positive pairs), (**C**) sensitivity $s_{post}$, and (**D**) variation of information $v$ as a function of complementarity determining region 3 (CDR3) length $l$ in the realistic synthetic dataset generated for this study. Solid lines represent the mean value averaged over five synthetic datasets. (**E–G**) Same as (**B–D**) but for the synthetic dataset from **Ralph and Matsen, 2022**, designed for the development and testing of the partis software. The solid lines represent the mean over the three datasets.

The online version of this article includes the following figure supplement(s) for figure 4:

**Figure supplement 1.** Performance of spectral SCOPer using V gene mutations.

**Figure supplement 2.** Performance of single-linkage clustering with fixed threshold.

Kleinstein, 2018). The SCOPer method using V and J gene mutations (**Nouri and Kleinstein, 2020**) was also tested, but gave worse results (**Figure 4—figure supplement 1**). Details about the used versions and parameters are referenced in the data availability section. We tested all algorithms on two synthetic datasets: a dataset simulated by the partis package and used in **Ralph and Matsen, 2022**, to benchmark partis against increasing levels of somatic hypermutations, and the synthetic data described above. That dataset is more realistic in the sense that it represents well the statistics of mutation patterns and, perhaps more importantly, the long-tail distribution of clone sizes observed in the data, with its large impact on the diversity of prevalences, which play an important role in the inference. The partis dataset is generated from a population genetics model. It provides a more independent test since it is not based on data used to develop the method and allows to study performance across different mutation rates.

First, we measure the inference time of each algorithm on our synthetic dataset. We find that the inference time is primarily affected by the size of the largest $VJl$ class. Therefore, we measure the inference time using the largest class found in donor 326651 of **Briney et al., 2019**, with the size of $N = 1.2 \times 10^5$ unique sequences. We then apply the methods to a series of subsamples of this class to get the computational time as a function of the subsample size (**Figure 4A**). We only allowed for runtimes below 1 hr. We find that only three methods achieve satisfactory performance (under an hour): the two methods introduced here, and the alignment-free method. The other two methods, SCOPer and partis, are limited to $VJl$ classes of small size ($< 10^4$ and $< 10^3$, respectively).

To compare the five algorithms in finite time, we test the accuracy of the methods using synthetic datasets with different CDR3 lengths, and with fixed mutation rate of 10% for the partis dataset (the mutation rate is not adjustable in our synthetic dataset as it mimics that of the data). We focus on short CDR3s, $l \in [15, 45]$, which are the most challenging for lineage inference. Clonal families with longer CDR3s are easy to reconstruct, and simple methods such as single-linkage clustering with a threshold

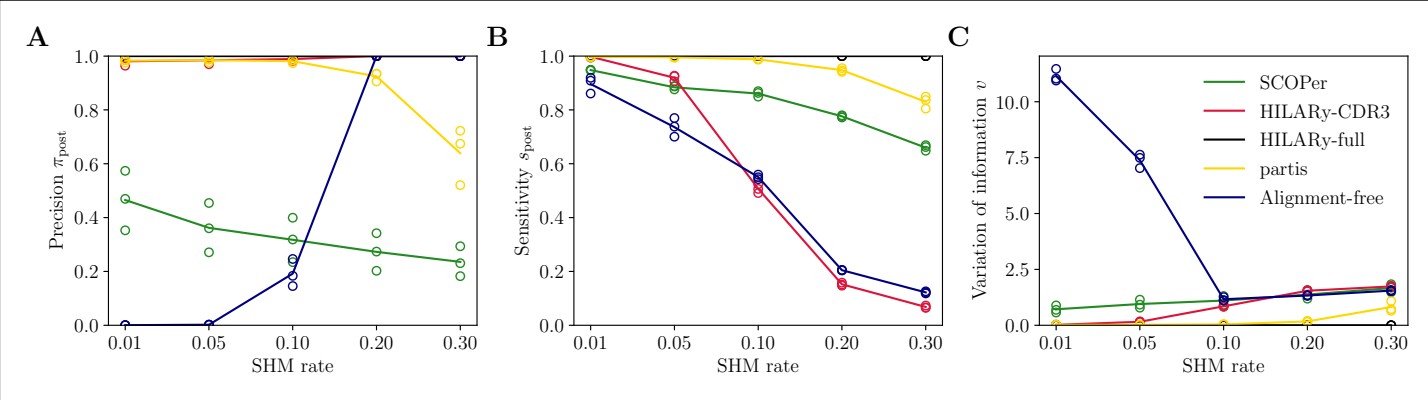

**Figure 5.** Performance of HILARy as a function of mutation rate for heavy chains, on synthetic data from *Ralph and Matsen, 2022*, designed for the development and testing of the partis software. (**A**) Clustering precision $\pi_{\text{post}}$ (post single-linkage clustering of positive pairs), (**B**) sensitivity $s_{\text{post}}$, and (**C**) variation of information $v$ as a function of mutation rate, using the heavy chain only. Solid lines represent the mean value averaged over the three datasets.

on mutational distance already work very well (*Figure 4—figure supplement 2A–C*). Each dataset contains $10^4$ unique sequences, so that the dominant VJ$l$ class is typically of size ~$10^3$ and can be handled by all five algorithms. We measure performance using three metrics applied to the resulting partition: pairwise sensitivity $s_{\text{post}}$ (*Figure 4B and E*), pairwise precision $\pi_{\text{post}}$ (*Figure 4C and F*), and the variation of information $v$ (*Figure 4D and G*). Performance measures as a function of mutation rate in the partis dataset are presented in *Figure 5*. Pairwise sensitivity $s_{\text{post}}$ and precision $\pi_{\text{post}}$ are a posteriori analogs of the a priori estimates defined before in *Equations 1 and 3*, now computed after propagating links through the transitivity rule of single-linkage clustering. Their value reflects not only the accuracy of the adaptive threshold but is also affected by the propagation of errors in single-flinkage clustering. Variation of information is a global metric of clustering performance which measures the loss of information from the true partition to the inferred one, and is equal to zero for perfect inference and positive otherwise (Methods).

Out of the five tested methods, only HILARy-full achieved both high sensitivity and high precision across all CDR3 lengths and for both synthetic datasets. HILARy-full is the only method reaching both high precision and sensitivity for CDR3s shorter than or equal to 30 nucleotides (*Figure 4B*), which corresponds to ~10% of a typical repertoire of productive IgGs (Figure 7A, inset).

The HILARy-CDR3 method achieves high precision everywhere by construction, but only reached good sensitivity for CDR3 lengths 24 and above. The alignment-free method also achieves high precision everywhere, but with low sensitivity, meaning that it erroneously breaks up clonal families into smaller subsets. These three methods achieve good precision, thanks to the use of a null model for the negative pairs. On the contrary, SCOPer has excellent sensitivity everywhere but only achieves high precision for large lengths ($l > 30$), suggesting that it erroneously merges short-CDR3 clonal families. Likewise, partis has high sensitivity but loses precision for short CDR3 on our realistic dataset, meaning that many clonal families are erroneously merged again. Note that our definition of precision and sensitivity differs from those used in *Ralph and Matsen, 2022*, which explains the differences between the performance measures reported here and in *Ralph and Matsen, 2022*. On the synthetic datasets from *Ralph and Matsen, 2022*, HILARy-full is the only method achieving high precision and sensitivity across mutation rates (*Figure 5*).

The variation of information offers a useful summary of the performance (*Figure 4D and G* and *Figure 5C*). According to that measure, only HILARy-full performs well across CDR3 lengths, mutation rates, and datasets. In particular, HILARy presents a clear advantage for challenging regions of the parameter space, such as CDR3 lengths below 30 nucleotides, and mutation rates of 20% and more.

For a typical repertoire, performance can be summarized into a global score by averaging over all CDR3 lengths in proportion of their abundance (assuming inference is perfect for CDR3 lengths larger than 45 regardless of the method). In this task, HILARy-full achieved 99.9% precision and 98.5% sensitivity; partis 93% and 96.9%; and SCOPer 96.6% and 99.1%. These scores are high because only

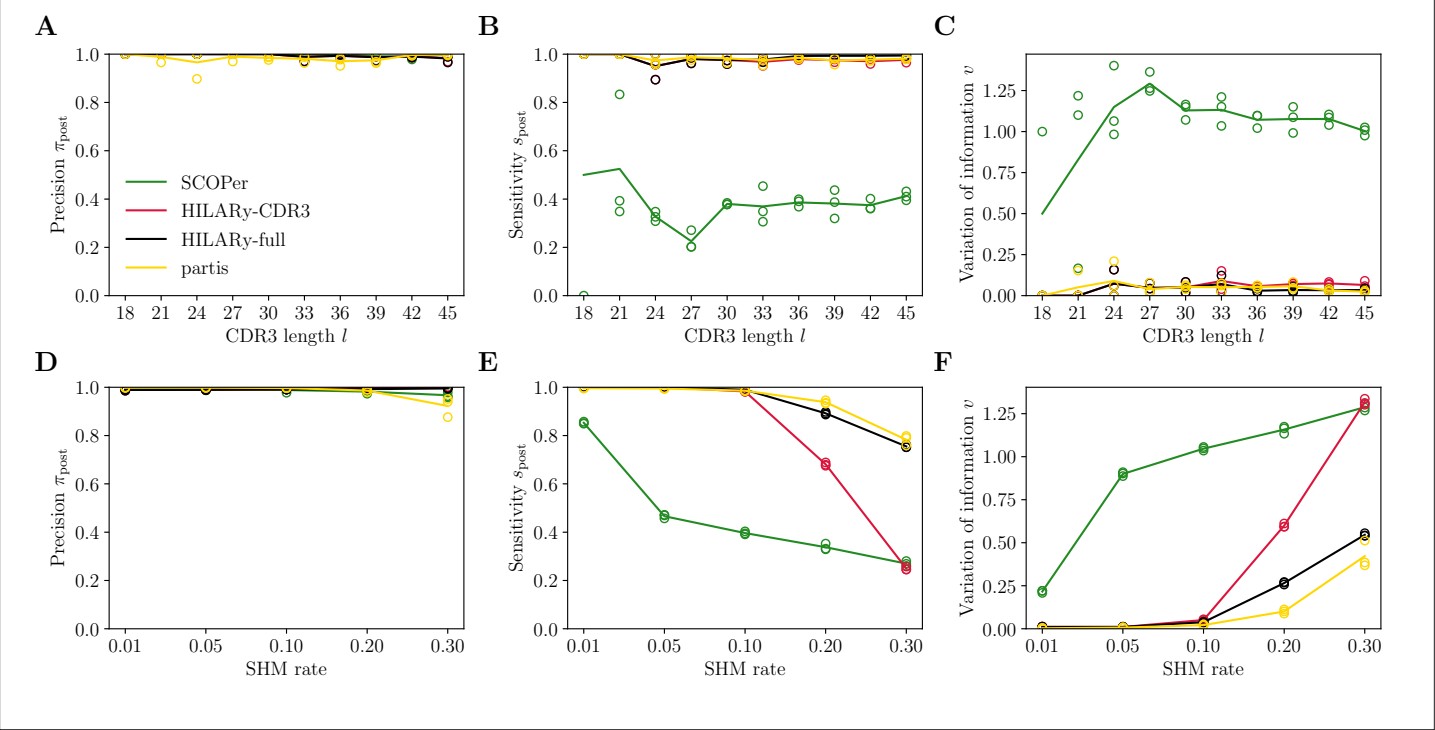

**Figure 6.** Benchmark of HILARy with paired light and heavy chains. (**A**) Clustering precision $\pi_{\text{post}}$ (post single-linkage clustering of positive pairs), (**B**) sensitivity $s_{\text{post}}$, and (**C**) variation of information $v$ as a function of complementarity determining region 3 (CDR3) length $l$, on the synthetic datasets from *Ralph and Matsen, 2022*, designed for the development and testing of the partis software. (**D–F**): Same as (**A–C**) but as a function of mutation rate.

a minority of lineages are difficult to infer. Nonetheless, HILARy-full provides a substantial gain in precision, while SCOPer presents a slight advantage in sensitivity.

We conclude that HILARy-CDR3 should be chosen for its consistently high sensitivity, specificity, and speed. In the case of the largest datasets, the faster HILARy-CDR3 is a useful alternative for long enough CDR3s in realistic repertoires.

## Extension to heavy- and light-chain paired data

We added an extension of HILARy to infer lineages from paired-chain repertoires, i.e., with paired light- and heavy-chain sequences. To extend HILARy-CDR3, we generalize the VJ$l$ class to a $V_H J_H V_L J_L l_{H+L}$ class, using the V and J genes from both the heavy and light chains, and the sum of their CDR3 lengths $l_{H+L} = l_H + l_L$. We then apply HILARy-CDR3 using the sum of the Hamming distances between the heavy- and light-chain CDR3s, normalized by $l_H + l_L$, as our new paired-chain $x$. The null distribution used is computed with soNNia using a default generation model for paired heavy and light chains. We incorporate the phylogenetic signal of both chains by concatenating their respective template genes, to obtain the total mutation counts $n_0 = n_{0,H} + n_{0,L}$, and using $L$ as the sum of the lengths of the $V_H$ and $V_L$ genes.

In *Figure 6A–C* we compare our method to SCOPer and partis on the synthetic dataset from *Ralph and Matsen, 2022*, as a function of CDR3 length, as our method for generating synthetic sequences could not be easily extended to add random light chains. Performance comparison as a function of mutation rate is presented in *Figure 6D–F*. HILARy performs better than SCOPer and comparably to partis, which was designed and tested against this dataset.

## Inference of clonal families in a healthy repertoire

We next use our method to infer the clonal families of the heavy-chain IgG repertoires of healthy donors from *Briney et al., 2019*. *Figure 7* summarizes key properties of the inferred clonal families of donor 326651. We take advantage of the consistency of our method across CDR3 lengths, as

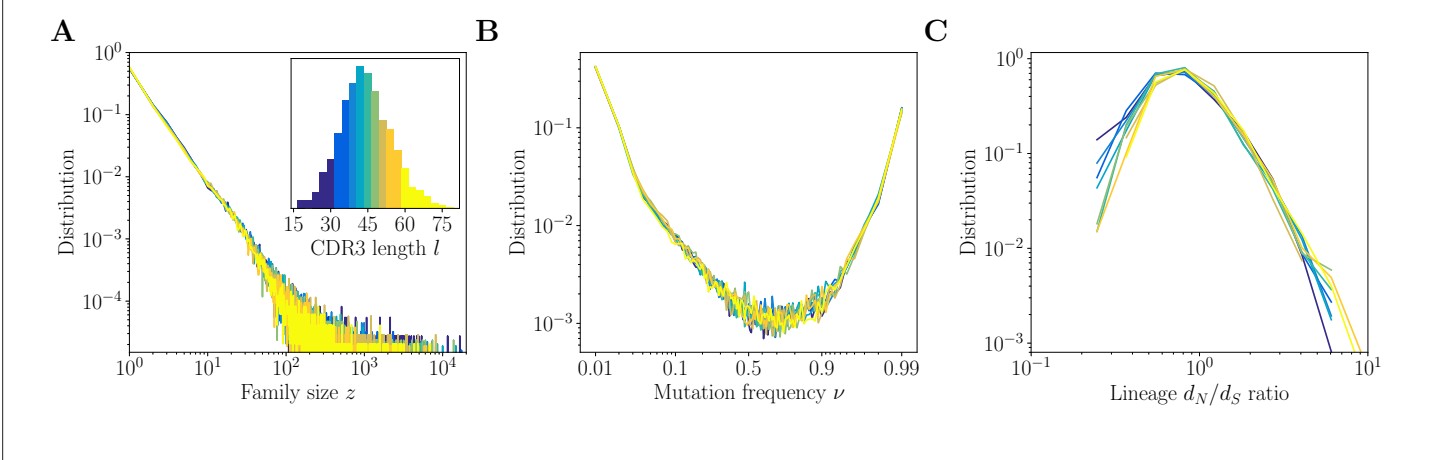

**Figure 7.** Inference results across complementarity determining region 3 (CDR3) lengths. Inference results for donor 326651 of **Briney et al., 2019**, are presented for nine quantiles of the CDR3 distribution, each containing between 8% and 12% of the total number of sequences (corresponding to nine colors in the inset of A). (**A**) Distributions of family size $z$. All CDR3 length quantiles exhibit universal power-law scaling with exponent −2.3. (**B**) Site frequency spectra estimated for families of sizes $z = 100$. Families of larger sizes were subsampled to $z = 100$ to subtract the influence of varying family sizes. (**C**) Distribution of lineage $d_N/d_S$ ratios computed for polymorphisms in CDR3 regions over all lineages within each nine quantile.

evidenced by the benchmark, to study how the lineage structure changes with the CDR3 length. To this end, we divide the dataset into nine quantiles, each containing ~10% of the total number of sequences (**Figure 7A**, inset).

We find that across the nine subsets of the data, the statistics of the lineage structure inferred with the mutations-based method are largely universal. The distribution of the clonal family sizes $z$ (**Figure 7A**) follows a power law across all CDR3 lengths under study, with no significant differences between different lengths. This results generalizes an earlier observation used above for generating synthetic datasets, but which was restricted to high-$\hat{\rho}$, high-$l$ V$Jl$ classes, and justifies a posteriori the use of a universal power law in the generative model.

For the largest families, of size $z \geq 100$, we compute two intra-lineage summary statistics: the site frequency spectrum, which gives the distribution of frequencies of point mutations within lineages, and the distribution of $d_N/d_S$ ratios between non-synonymous and synonymous CDR3 polymorphisms within clonal families (estimated by counting). To avoid the bias of the varying family sizes, we subsampled all families to size $z = 100$.

Under models of neutral evolution with fixed population size, the distribution of point-mutation frequencies $\nu$ goes as $\nu^{-1}$. Here, we observe a non-neutral profile of the spectrum, with an upturn at large allele frequencies $\nu > 0.5$ (**Figure 7B**). It is a known signature of selection or of rapid clonal expansion (**Horns et al., 2019**; **Nourmohammad et al., 2019**). We find that site frequency spectra are universal for all CDR3 lengths, suggesting that the dynamics that give rise to the structure of lineages and the subsequent dynamics that influence the sampling of family members do not depend on the CDR3 length.

The lineage $d_N/d_S$ ratio is also largely consistent across CDR3 lengths (**Figure 7C**), while spanning two orders of magnitude, suggesting a wide gamut of selection forces. We could have expected longer loops to be under stronger purifying selection (lower $d_N/d_S$) to maintain their specificity and folding. Instead, we observe that short CDR3s have more lineages with low $d_N/d_S$. This may be due to different sequence context and codon composition in short versus long CDR3s. Short junctions are largely templated, whereas long junctions have long, non-templated insertions, and it was shown that templated regions have evolved their codons to minimize the possibility of non-synonymous mutations (**Saini and Hershberg, 2015**), which would lead to a lower $d_N/d_S$, regardless of selection.

## Discussion

Clonal families are the building blocks of memory repertoire shaped by VDJ recombination and subsequent somatic hypermutations and selection. Repertoire sequencing datasets enable new approaches

to understand these processes. They allow us to model the different sources of diversity and measure the selection pressures involved. To take full advantage of this opportunity, we need to reliably identify independent lineages.

Here, we introduced a general framework for studying the methods for partitioning high-throughput sequencing of BCR repertoire datasets into clonal families. We have identified the main factors that influence the difficulty of this inference task: low clonality levels and short recombination junctions. We quantified the clonality level using the definition of pairwise prevalence $\rho$ and introduced a method to estimate it a priori, without knowing the partition. We found the prevalence levels across $VJl$ classes to span three orders of magnitude (*Figure 2B*), unraveling the varying degree of complexity.

We leveraged the soNNia model of VDJ recombination to quantify the CDR3 diversity and constructed a null expectation for the divergence of independent recombination products. This null model enabled the design of a CDR3-based clustering method with an adaptive threshold, HILARy-CDR3, that allows us to keep the precision of inference high across prevalences and CDR3 lengths. Owing to the prefix tree representation of the CDR3 sequences, this method is characterized by very short inference times, thanks to avoiding all pairwise comparisons in single-linkage clustering. As expected, we found that the adaptive threshold choice limits the sensitivity of inference in the regime of short junctions and low prevalence (*Figure 3F*, below the black line).

To remedy the limitations of the CDR3-based approach, we developed a mutation-based method (HILARy-full). We found that including the phylogenetic signal of shared mutations in highly mutated sequences allows us to properly classify them into lineages despite significant CDR3 divergence. We studied the performance of the method using synthetic data and found significant improvement with respect to HILARy-CDR3: we extended the range of high-precision and high-sensitivity performance to cover all values of prevalence and CDR3 lengths observed in productive data (*Figure 3G*).

We have compared the two methods developed here with state-of-the-art approaches: the partis (*Ralph and Matsen, 2016*) and SCOPer (*Nouri and Kleinstein, 2018*) algorithms, and the alignment-free method (*Lindenbaum et al., 2021*). Compared to these methods, HILARy relies on a probabilistic model of VDJ recombination and selection, which allows it to explicitly control for precision. This is not possible in partis, which relies on likelihood ratio test to merge candidate clusters together to form families. SCOPer also chooses a clustering threshold based on the pairwise distribution of distances, but without a null model. Another innovation of HILARy-full is to use a null expectation for the number of shared mutations. This feature makes the method robust to varying levels of mutation rates across sequences. HILARy achieves optimal efficiency by combining CDR3-based and mutation-based information. Typically, a large part of the dataset doesn't require the use of the full method, allowing for greatly reduced inference times. HILARy relies on the soNNia model, which is based on a neural network, and benefits from its expressivity to quantify the purifying selection that modifies the VDJ recombination statistics. We found the performance of this model satisfactory when applied to healthy memory repertoires, in agreement with previous findings (*Isacchini et al., 2021*; *Ruiz Ortega et al., 2023*). For subsets of the repertoire with less challenging characteristics, such as low mutation rates, long CDR3s, or high pairwise prevalence $\rho$, simpler methods can effectively reconstruct clonal families with high precision and sensitivity. As demonstrated in *Balashova et al., 2024*, single-linkage clustering outperforms state-of-the-art approaches for simulated samples based on real datasets with mutation rates ranging between 1.3% and 5.5%. As part of our clonal inference package, we provide our own implementation of single-linkage clustering based on mutational distance, which leverages a prefix tree representation method to speed up inference. We found this approach to be comparable to HILARy for long CDRs and low mutation rates (*Figure 4—figure supplement 2*).

Purifying selection is expected to be more pronounced in datasets of disease-specific cohorts and a default soNNia model may overestimate the diversity (*Mayer and Callan, 2022*) and lead to underestimation of the fallout rate. The inference framework introduced here could still be applied with more sophisticated models of selection, and take advantage of higher levels of clonality that characterize many disease-specific datasets (*Nielsen et al., 2020*; *Turner et al., 2020*).

We applied the mutations-based method to infer lineages in a repertoire of a healthy donor, sequenced at great depth (*Briney et al., 2019*). We took advantage of the consistency our method exhibits across CDR3 lengths to find that the statistics of lineages, including a heavy-tail distribution of family sizes as well as signatures of selection, are universal and independent of the CDR3 length. This result implies that the dynamics of expansion, mutation, and selection are independent of the

CDR3 and suggests they are dictated by the rules of affinity maturation and memory formation rather than BCR specificity. It advocates for the use of RNA sequencing data to quantify these general principles (*Mayer and Callan, 2022*; *Hoehn et al., 2019*). Identifying clonal families with high accuracy is paramount in such approaches as it avoids the potential biases of different family sizes and varying levels of clonality.

The algorithm for clonal family identification presented here is a robust inference method that enables a reliable partition of a memory B-cell repertoire into independent lineages. Using synthetic datasets we demonstrated it is distinguished by consistently high precision and high sensitivity across different junction lengths and levels of clonality, while very fast compared to previous methods. It is therefore a useful tool to explore the diversity of the repertoires and improves our ability to interpret repertoire sequencing datasets.

## Methods

### Data preprocessing and alignment

We focus the analysis high-throughput RNA sequencing data of IgH-coding genes (*Briney et al., 2019*). The sequences were barcoded with unique molecular identifiers (UMIs) to correct for the PCR amplification bias and correct sequencing errors. We aligned raw sequences using presto of the Immcantation pipeline (*Vander Heiden et al., 2014*) with tools allowing for correcting errors in UMIs and deal with insufficient UMI diversity. Reads were filtered for quality and paired using default presto parameters. We selected only sequences aligned with the IgG primer and therefore the lineage analysis is limited to the IgG subset of the repertoire. Preprocessed data was then aligned to V, D, and J templates from IMGT (*Giudicelli et al., 2006*) database using IgBlast (*Ye et al., 2013*). After processing, all UMI count information is discarded and only unique nucleotide sequences are kept for further analysis.

Pairs of sequences stemming from the same VDJ recombination are expected to have the same CDR3 length $l$ and align to the same V and J templates. An exception could be caused by a insertion or deletion within the CDR3 that would alter its length as a result of the somatic hypermutation process. Such indel events are rare and generally selected against (*Lupo et al., 2022*), therefore in what follows we shall assume the effect of these events is negligible. The inference could also be affected by the misalignment of either V or J templates but we previously found the effect of alignment errors to be insignificant for identifying VJ classes (*Spisak et al., 2020*) (the alignment of the D template is error-prone and unreliable, hence not used in the inference procedure). Importantly, the two simplifications described here would result in decreased sensitivity of inference but are not expected to affect its precision.

### Modeling junctional diversity

The extraordinary diversity of VDJ rearrangements can be efficiently described and quantified using probabilistic models of the recombination process as well as subsequent purifying selection. Sequence-based models can assign to each receptor sequence $s$, its total probability of generation, $P_{gen}(s)$ (*Murugan et al., 2012*; *Elhanati et al., 2015*; *Marcou et al., 2018*) as well as a selection factor $Q(s)$, inferred so as to match frequencies $P_{data}(s)$ of the sequences with a model-based distribution (*Elhanati et al., 2014*; *Sethna et al., 2020*; *Isacchini et al., 2021*)

$$P_{post}(s) = Q(s)P_{gen}(s). \tag{5}$$

The $P_{gen}$ model was inferred using unmutated out-of-frame sequences from *Briney et al., 2019*, using the IGoR software (*Marcou et al., 2018*). The selection function $Q$ model was learned using unmutated productive IgM sequences from *Briney et al., 2019*, using soNNia software (*Isacchini et al., 2021*).

The post-selection distribution $P_{post}$ describes the diversity of the CDR3 regions and in doing so provides an expectation of pairwise distances between unrelated, independently generated sequences of same length $l$ (*Isacchini et al., 2021*). As the soNNia software does not include somatic hypermutations, the underlying assumption is that additional diversity on the CDR3 caused by hypermutations doesn't affect the distribution of pairwise distances. This assumption is justified by the quality of the fit. We can define

$$P_{\text{F}}(n|l) = \left\langle \delta_{|s_1 - s_2|, n} \right\rangle_{s_1, s_2 \sim P_{\text{post}}(\cdot|l)}, \tag{6}$$

where $|s_1 - s_2|$ stands for (Hamming) distance between sequences $s_1$ and $s_2$. This definition of the null distribution is a straightforward recipe for its estimation using (Monte Carlo) samples from $P_{\text{post}}$.

Should $P_{\text{post}}$ differ significantly from the empirical frequencies $P_{\text{data}}$ one can resolve to the following alternative

$$P'_{\text{F}}(n|l) = \left\langle \delta_{|s_1 - s_2|, n} \right\rangle_{s_1 \sim P_{post}(\cdot, l), s_2 \sim P_{data}(\cdot|l)}, \tag{7}$$

the equivalent of the negation distribution as defined in *Lindenbaum et al., 2021*, and used in our evaluation of the alignment-free method (*Lindenbaum et al., 2021*) in the method benchmark analysis.

## Estimation of pairwise prevalence

Pairwise prevalence is defined as the ratio of pairs of related sequences to the total number of pairs of sequences in a given set. Related sequences share an ancestor and have diverged by independent somatic mutations, post-recombination. Low prevalence can be a major difficulty for any inference procedure as any misassignment (or fallout) will result in a drastic loss of sensitivity or precision. It is instrumental to have an a priori estimate of pairwise prevalence before the families are identified.

To estimate the prevalence from the distribution of distances $P(n)$ for a given set of sequences (typically a VJ$l$ class or $l$ class), we propose the following expectation-maximization procedure. We stipulate the distribution in question is a mixture distribution of two components, $P_{\text{F}}(n)$, the expectation for unrelated sequences defined as above, and $P_{\text{T}}(n)$, describing related sequences, modeled using a Poisson distribution

$$P_{\text{T}}(n) = \frac{(\mu l)^n}{n!} e^{-\mu l}, \tag{8}$$

where $\mu$ is the mean divergence per base pair. If a particular CDR3 length $l$ is represented by unusually large number of VJ$l$ classes, the resultant shape of the positive distribution is often closer to a geometric profile, and is then modeled using $P_{\text{T}}(n) = (1 - M)M^n$, where $M = \frac{1}{1 + \mu l}$. In sum

$$P(n) = \rho P_{\text{T}}(n) + (1 - \rho)P_{\text{F}}(n). \tag{9}$$

In a standard fashion, we proceed iteratively by calculating the expected value of the log-likelihood (pairs of sequences indexed by $i$)

$$Q(\rho, \mu|\rho_t, \mu_t) = \sum_i P_t(i \in \text{T}) \log P_{\text{T}}(n_i|\mu) + P_t(i \in \text{F}) \log P_{\text{F}}(n_i), \tag{10}$$

where the membership probabilities are defined as

$$P_t(i \in \text{T}) = P(i \in \text{T}|n_i, \mu_t, \rho_t) \tag{11}$$

$$= \frac{\rho_t P_{\text{T}}(x|\mu_t)}{\rho_t P_{\text{T}}(x|\mu_t) + (1 - \rho_t)P_{\text{F}}(x)} \tag{12}$$

$$P_t(i \in \text{F}) = P(i \in \text{F}|n_i, \mu_0, \rho_0) = 1 - P_t(i \in \text{T}). \tag{13}$$

We then find the maximum

$$\mu_{t+1}, \rho_{t+1} = \text{argmax } Q(\rho, \mu|\rho_0, \mu_0) \tag{14}$$

and iterate the expectation and maximization steps until convergence, $|\rho_{t+1} - \rho_t| < \epsilon$, to obtain $\hat{\rho} = \rho_{t+1}$.

Results for largest VJ$l$ class within each $l$ class can be found in *Figure 2—figure supplement 3* and results for $l$ classes using a geometric distribution can be found in *Figure 2—figure supplement 6*. Dependence of maximum likelihood prevalence estimates $\hat{\rho}$ on class size $N$ is plotted in *Figure 2—figure supplement 7*.

## HILARy-CDR3

The standard method for CDR3-based inference of lineages proceeds through single-linkage clustering with a fixed threshold on normalized Hamming distance divergence (fraction of differing nucleotides) (*Kepler, 2013*; *Uduman et al., 2014*; *Yaari and Kleinstein, 2015*; *Nourmohammad et al., 2019*). This crude method suffers from inaccuracy as it loses precision in the case of highly mutated sequences and junctions of short length (see *Figure 4—figure supplement 2*). If junctions are stored in a prefix tree data structure (*Knuth, 2013*) single-linkage clustering can be performed without comparing all pairs and hence is typically orders of magnitude faster than alternatives. The prefix tree is a search tree constructed such that all children of a given node have a common prefix, the root of the tree corresponding to an empty string, and leaves corresponding to unique sequences to be clustered. To find neighbors of a given sequence it suffices to traverse the prefix tree from the corresponding leaf upward and compute the Hamming distance at branchings. This method limits the number of unnecessary comparisons and greatly improves the speed of Hamming distance-based clustering (*Boytsov, 2011*). We implement the prefix tree structure to accommodate CDR3 sequences. Briefly, all the CDR3 sequences of identical length are stored in the leaves of a prefix tree (*Navarro, 2001*; *Boytsov, 2011*), implemented as a quaternary tree where each edge is labeled by a nucleobase (A, T, C, or G). The neighbors of a specific sequence are found by traversing the tree from top to bottom, exploring only the branches that are under a given Hamming distance from the sequence. Clusters are obtained by iterating this procedure and removing all the neighbors from the prefix tree until no sequences remain. The package is coded in C++ with a Python interface and is available independently. The time performance of this method for high-sensitivity and high-specificity partitions is studied as a part of the method benchmark analysis.

We take advantage of the speed of a prefix tree-based clustering to perform single-linkage clustering. Besides the algorithmic speed-up afforded by the prefix tree, the difference with previous methods is that we use an adaptive threshold. For any dataset, we define two CDR3-based partitions, high-sensitivity and high-precision clustering, corresponding to two choices of threshold.

The high-precision partition is obtained by setting the threshold $t$ to $t^*_{\text{prec}}$ as the largest $t$ such $\hat{\pi}(t) \leq \pi^*$, with $\pi^* = 0.99$ (99% precision), where $\hat{\pi}(t)$ is given by *Equation 1–3*. To get the high-sensitivity partition, we set the threshold to $t^*_{\text{sens}}$, the smallest $t$ such that $\hat{s}(t) \geq s^*$, where $s^* = 0.9$ (90% sensitivity), where $\hat{s}(t)$ is given by *Equation 3*.

We apply these thresholds to the single-linkage clustering described above to generate the precise and sensitive partitions, which are then used by the mutations-based method to find an optimal partition that merges the fine clusters within the coarse clusters (Methods and *Figure 3—figure supplement 1*). We refer to the high-precision partition from the CDR3 alone as HILARy-CDR3, and the mutation-based method as HILARy-full.

Finally, the structure of families leads to propagation of errors that lowers the precision with respect to the a priori estimate $\hat{\pi}$. Denoting family size as $z$, one error accounted for in $\hat{\text{FP}}$ causes, on average, $\langle z \rangle^2 - 1$ extra errors by merging two families. If the a priori precision $\hat{\pi}$ is high, we can neglect the second order effect of these two families simultaneously affected by other $\hat{\text{FP}}$ pairs. Therefore the expected precision (*Equation 1*) of the resulting partition reads

$$\langle \pi_{\text{post}} \rangle \simeq \frac{1}{1 + (\langle z \rangle^2 - 1)(1 - \hat{\pi})} \tag{15}$$

where we assumed $\hat{s} \simeq 1$. For $\hat{\pi} = 99\%$ and $\langle z \rangle \simeq 2$ this formula gives $\langle \pi_{\text{post}} \rangle \simeq 97\%$.

## Synthetic data generation

To generate synthetic data we make use of the statistics of tree topologies of the lineages identified in the high-sensitivity and high-precision regime of CDR3-based inference from the data (yellow region above the black line in *Figure 3F*). We denote the set of these lineages by $\mathcal{L}$. We assume that to good approximation the mutational process and the selection forces that shaped the mutational landscape in these lineages do not depend on the CDR3 length.

To test the performance of different inference methods across CDR3 lengths, we build synthetic datasets of fixed length.

In the first step, we choose the number of families $N$. We then draw $N$ independent family sizes from the family size distribution of the form observed in healthy datasets

$$p(z) = \frac{z^{-\alpha}}{Z_\alpha}, \tag{16}$$

where $Z_\alpha = \sum_{z \geq 1} z^{-\alpha} = \zeta(\alpha, 1)$. In the next step, we assign a naive progenitor to each lineage by sampling from the $P_{\text{post}}$ distribution, selecting sequences with a prescribed length $l$ (**Figure 2—figure supplement 4**). We then choose a lineage in the set of reconstructed lineages $\mathcal{L}$ at random among lineages of size $z$ (or, for large sizes, the lineage of the closest size smaller than $z$). To create a lineage with the same mutation patterns as the real data, we then identify all unique mutations in the lineage from $\mathcal{L}$ using standard alignment and tree reconstruction methods described in **Spisak et al., 2020**, and for each mutation denote the labels of members of the lineage that carry it. For each mutation, this defines a configuration of labels, one of $2^z - 1$ possible. We subsequently loop through observed configurations and choose new positions for all mutations to apply them to the synthetic progenitors of the ancestor, using the position- and context-dependent model of **Spisak et al., 2020**. The number of mutations assigned to a given configuration is rescaled by a factor $\frac{L + l}{L_0}$ where $L$ is the templated length of the synthetic ancestral sequence and $L_0$ is the templated length of the model lineage from $\mathcal{L}$.

This way a synthetic lineage preserves all properties of the lineages of long CDR3s found in the data, particularly the mutational spectra (**Figure 2—figure supplement 5**) except for the ancestral sequences and the identity of mutations.

## HILARy-full

We compute the expected distributions of the CDR3 Hamming distance $n$, and the number of shared mutations $n_0$, under a uniform mutation rate assumption. In other words, we assume that the probability that a given position was mutated, given a mutation happened somewhere in a sequence of length $L$, equals $L^{-1}$ (we know this not to be true, see, e.g., **Spisak et al., 2020**, but it allows for simple computations). It follows that the probability that a given position has not mutated once in a series of $n$ mutations is $(1 - L^{-1})^n$.

### Expectation of $n_0$ under the null hypothesis

For $n_0$ shared mutations, under the null hypothesis (we operate under the null hypothesis here since otherwise to estimate $n_0$ we would need to make assumptions about the law that governs B-cell phylogeny topologies), the likelihood reads

$$P_{\text{F}}(n_0|n_1, n_2, L) = \binom{L}{n_0} \text{p}^{n_0}(1 - \text{p})^{L - n_0}, \tag{17}$$

where the probability that the same position independently mutated in series of $n_1$ and $n_2$ mutations is

$$\text{p} = \left(1 - (1 - L^{-1})^{n_1}\right)\left(1 - (1 - L^{-1})^{n_2}\right). \tag{18}$$

In the limit of large $L$, we have at leading order

$$\text{p} = \frac{n_1 n_2}{L^2}, \tag{19}$$

$$
\begin{aligned}
P_{\text{F}}(n_0|n_1, n_2, L) &\simeq \binom{L}{n_0}\left(\frac{n_1 n_2}{L^2}\right)^{n_0}\left(1 - \frac{n_1 n_2}{L^2}\right)^{L - n_0} \\
&\simeq \frac{\left(\frac{n_1 n_2}{L}\right)^{n_0}}{n_0!} e^{-\frac{n_1 n_2}{L}},
\end{aligned}
\tag{20}
$$

where the last approximation assumes $n_1 n_2 \ll L^2$, which holds when mutation rates are small. Therefore, $P_{\text{F}}(n_0|n_1, n_2, L)$ may be approximated by a Poisson distribution of parameter $\frac{n_1 n_2}{L}$, yielding:

$$\langle n_0 \rangle_{\text{F}} \simeq \frac{n_1 n_2}{L}, \quad \sigma_{\text{F}}(n_0) \simeq \sqrt{\frac{n_1 n_2}{L}}. \tag{21}$$

## Expectation of $n$ under the hypothesis of related sequences

The $n$ divergence of two CDR3s is interpreted as divergent mutations under the hypothesis that $s_1$ and $s_2$ are related. These mutations were harbored in parallel with $n_L = n_1 + n_2 - 2n_0$ mutations that occurred in the templated regions ($n_0$ mutations arrived before the divergence of the two sequences began).

Under the assumption of a uniform mutation rate, the $n_L$ mutations inform the prediction of the number of mutations expected in the CDR3. Indeed, they are related through a hidden variable, the expected number of mutations per base pair, denoted $\mu$. Integrating over this quantity we obtain

$$P_\text{T}(n|n_L, l, L) = \int_0^\infty d\mu\, P_\text{T}(n|\mu, l) P_\text{T}(\mu|n_L, L), \tag{22}$$

where we convolute the positive distribution (**Equation 8**),

$$P_\text{T}(n|\mu, l) = \frac{(\mu l)^n}{n!} e^{\mu l} \tag{23}$$

and, using the Bayes rule under uniform prior over $\mu$,

$$P_\text{T}(\mu|n_L, L) = L^{-1} P_\text{T}(n_L|\mu, L) = \frac{(\mu L)^{n_L}}{n_L! L} e^{\mu L}. \tag{24}$$

The result is a negative binomial distribution,

$$P_\text{T}(n|n_L, l, L) = \left(\frac{L}{l+L}\right)^{n_L+1} \left(\frac{l}{l+L}\right)^n \binom{n+n_L}{n}, \tag{25}$$

with

$$\langle n \rangle_\text{T} = \frac{l}{L}\left(n_L + 1\right), \quad \sigma_\text{T}(n) = \frac{1}{L}\sqrt{l(l+L)(n_L+1)}. \tag{26}$$

## Merging fine-partition clusters

HILARy-full relies on the results (**Equation 26**) and (**Equation 21**) to define the rescaled variables (**Equation 4**)

$$x' = \frac{n - \langle n \rangle_\text{T}}{\sigma_\text{T}(n)}, \quad y = \frac{n_0 - \langle n_0 \rangle_\text{F}}{\sigma_\text{F}(n_0)}. \tag{27}$$

We expect $y \approx 0$, $x' > 0$ for unrelated sequences, and $x' \approx 0$, $y > 0$ for related sequences. So we expect $x' - y > 0$ for unrelated sequences, and $x' - y < 0$ for related sequences. We use $x' - y$ as a distance for single-linkage clustering, with adaptive threshold to control performance. The threshold $t'$ is chosen to achieve a desired precision of $\pi^* = 0.99$ as in HILARy-CDR3. To this end we use soNNia-based estimate of null distribution $P_F(n|l)$ (**Equation 6**), the data-derived distribution of the number of mutations, $P(n_1)$, and further assume $n_0 \sim \frac{n_1 n_2}{L}$ to compute the null distribution $P_F(x' - y|l)$. We can now choose a target $\pi^*$ and compute $t'$ such that $\hat{\pi}(t') = \pi^* = 0.99$ using **Equations 1–3**, the prevalence $\hat{\rho}$ inferred as explained earlier in the CDR3-based method, and assuming $\hat{s} \simeq 1$. As the computation of $t'$ depends on the inferred prevalence, we use this procedure only for VJ$l$ classes with enough sequences for a reliable $\hat{\rho}$ (**Figure 3—figure supplement 2**), namely for sizes larger than 100. For smaller sizes the threshold was set to the default value of 0.

To reduce the number of pairwise computations, we do not apply single-linkage clustering directly, but instead merge fine-partition clusters within coarse-partition clusters, where the fine and coarse partitions were previously obtained using the CDR3-based method (see section HILARy-CDR3). Specifically, we compute $x' - y$ for all pairs of sequences that belong to the same coarse cluster, but to different fine clusters. Two fine-partition clusters are then merged if there exist any two sequences belonging to each of the two clusters for which $x' - y < t'$. Note that this is equivalent to performing single-linkage clustering on all sequences using the distance $-\infty$ for pairs inside a precise cluster and $x' - y$ otherwise.

## Evaluation methods

In this section, we introduce the variation of information $v$, used for evaluating alternative methods for clonal family inference in the benchmark analysis. It is a useful summary statistic to quantify the performance of inference as it is affected by its precision as well as sensitivity (*Brown et al., 2007*). Variation of information $v(r, r^*)$ measures the information loss from the true partition $r^*$ to the inference result $r$ (*Zurek, 1989*; *Meilă, 2003*). To define the variation of information we first introduce the entropy $S(r)$ of a partition $r$ of $N$ sequences into clusters $c$ as

$$S(r) = -\sum_{c \in r} \frac{n(c)}{N} \log \frac{n(c)}{N}, \tag{28}$$

where $n(c)$ denotes the number of sequences in cluster $c$. The mutual information between two partitions $r$ and $r^*$ can then be computed as

$$I(r, r^*) = \sum_{c \in r} \sum_{c^* \in r^*} \frac{n(c, c^*)}{N} \log \frac{n(c, c^*)}{N}, \tag{29}$$

where $n(c, c^*)$ denotes the number of overlapping elements between cluster $c$ in partition $r$ and cluster $c^*$ in partition $r^*$. Finally, variation of information is given by

$$v(r, r^*) = S(r) + S(r^*) - 2I(r, r^*). \tag{30}$$

Variation of information is a metric in the space of possible partitions since it is non-negative, $v(r, r^*) \geq 0$, symmetric, $v(r, r^*) = v(r^*, r)$, and obeys the triangle inequality, $v(r_1, r_3) \leq v(r_1, r_2) + v(r_2, r_3)$ for any three partitions (*Zurek, 1989*).

## Code and data availability

We used version 1.2.0 for spectral SCOPer, 1.3.0 for SCOPer using the V and J mutation presented in *Figure 4—figure supplement 1*, version 1.2.0 for HILARy, version 0.16.0 for partis, and the code from this repository https://bitbucket.org/kleinstein/projects/src/master/Lindenbaum2020/Example.ipynb for the alignment-free method. The HILARy tool with Python implementations of the CDR3 and mutation-based methods introduced above can be found at https://github.com/statbiophys/HILARy (copy archived at *Athènes, 2024*). The standalone prefix tree implementation can be found at https://github.com/statbiophys/ATrieGC (copy archived at *Dupic, 2024*). A complete guide to our benchmark procedure can be found in the README of the folder https://github.com/statbiophys/HILARy/tree/main/data_with_scripts where we make available scripts to infer lineages and reproduce the benchmark figures of this article. We also upload this folder with all input and output data at https://zenodo.org/records/10676371.

# Acknowledgements

The study was supported by European Research Council COG 724208 and ANR-19-CE45-0018 'RESP-REP' from the Agence Nationale de la Recherche grants and DFG grant CRC 1310 'Predictability in Evolution'.

# Additional information

### Competing interests

Aleksandra M Walczak: Senior editor, eLife. The other authors declare that no competing interests exist.

## Funding

| Funder | Grant reference number | Author |
|---|---|---|
| European Research Council | COG 724208 | Natanael Spisak<br>Thomas Dupic<br>Thierry Mora<br>Aleksandra M Walczak |
| Agence Nationale de la Recherche | ANR-19-CE45-0018 `RESP-REP' | Natanael Spisak<br>Thomas Dupic<br>Thierry Mora<br>Aleksandra M Walczak |
| Deutsche Forschungsgemeinschaft | CRC 1310 `Predictability in Evolution'. | Natanael Spisak<br>Gabriel Athènes<br>Thomas Dupic<br>Thierry Mora<br>Aleksandra M Walczak |

The funders had no role in study design, data collection and interpretation, or the decision to submit the work for publication.

## Author contributions

Natanael Spisak, Conceptualization, Resources, Data curation, Software, Formal analysis, Validation, Investigation, Visualization, Methodology, Writing – original draft, Writing – review and editing; Gabriel Athènes, Resources, Data curation, Software, Validation, Investigation, Visualization, Methodology, Writing – review and editing; Thomas Dupic, Resources, Software, Methodology; Thierry Mora, Aleksandra M Walczak, Conceptualization, Resources, Formal analysis, Supervision, Funding acquisition, Investigation, Methodology, Writing – original draft, Project administration, Writing – review and editing

## Author ORCIDs

Natanael Spisak http://orcid.org/0000-0002-6332-047X
Gabriel Athènes http://orcid.org/0009-0007-4668-884X
Thierry Mora https://orcid.org/0000-0002-5456-9361
Aleksandra M Walczak https://orcid.org/0000-0002-2686-5702

## Decision letter and Author response

Decision letter https://doi.org/10.7554/eLife.86181.sa1
Author response https://doi.org/10.7554/eLife.86181.sa2

---

# Additional files

## Supplementary files

• MDAR checklist

## Data availability

The current manuscript is a computational study, so no data have been generated for this manuscript. All data used is publicly available. The HILARy tool with Python implementations of the CDR3 and mutations-based methods introduced above can be found at https://github.com/statbiophys/HILARy (copy archived at *Athènes, 2024*). The standalone prefix tree implementation can be found at https://github.com/statbiophys/ATrieGC (copy archived at *Dupic, 2024*). A complete guide to our benchmark procedure can be found in the README of the folder https://github.com/statbiophys/HILARy/tree/main/data_with_scripts where we make available scripts to infer lineages and reproduce the benchmark figures of this article. We also upload this folder with all input and output data at https://zenodo.org/records/10676371.

The following dataset was generated:

| Author(s) | Year | Dataset title | Dataset URL | Database and Identifier |
|---|---|---|---|---|
| Spisak N, Athènes G, Athènes G, Dupic T, Mora T, Walczak A | 2024 | Combining mutation and recombination statistics to infer clonal families in antibody repertoires | https://doi.org/10.5281/zenodo.10676370 | Zenodo, 10.5281/zenodo.10676370 |

The following previously published dataset was used:

| Author(s) | Year | Dataset title | Dataset URL | Database and Identifier |
|---|---|---|---|---|
| Briney B, Inderbitzin A, Joyce C, Burton DR | 2019 | Uniqueness, commonality and exceptional diversity in the baseline human antibody repertoire | https://www.ncbi.nlm.nih.gov/bioproject/?term=PRJNA406949 | NCBI BioProject, PRJNA406949 |

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
