## [Editor Report]

This fundamental study provides a new, high-performance algorithm for B-cell clonal family inference. The new algorithm is highly innovative and based on a rigorous probabilistic analysis of the relevant biological processes and their imprint on the resulting sequences. The strength of evidence regarding the algorithm's performance is convincing, as the algorithm has been benchmarked against two state-of-the-art methods for clonal family inference on two synthetic data sets generated with two independent, state-of-the-art methods for B cell repertoire simulation. This work will be fundamental to immunologists and important to any researcher or clinician utilizing B cell receptor repertoires in their field.

---

## [Decision Letter]

**Decision letter after peer review:**

Thank you for submitting your article "Combining mutation and recombination statistics to infer clonal families in antibody repertoires" for consideration by *eLife*. Your article has been reviewed by 3 peer reviewers, one of whom is a member of our Board of Reviewing Editors, and the evaluation has been overseen by Miles Davenport as the Senior Editor. The following individuals involved in the review of your submission have agreed to reveal their identity: Frederick A Matsen (Reviewer #2); Kenneth B Hoehn (Reviewer #3).

Essential revisions:

1) Provide clarification regarding how different data sets (synthetic versus real) were used for different steps during algorithm development and validation, perhaps by including a flow chart of inputs/outputs/dependencies.

2) Conduct an evaluation that is not circular, such as using additional simulated data sets beyond the one based on the same data used to develop the method.

3) Provide details about the versions and settings of the competitor programs included in the benchmarking, as well as other general information such as: what type of machine was used? Was the use of multiple threads enabled for all programs? When multiple methods are available within a single program (e.g., scoper), indicate which were used and justify the choice. Etc.

4) Make the synthetic data publicly available as well as the scripts used for the benchmarking analysis.

5) Define "precision" prominently and use it in that sense consistently throughout the manuscript.

*Reviewer #1 (Recommendations for the authors):*

Regarding point 1 in the public review, examples of more specific questions along these lines are included below. A flow chart figure showing the steps, data inputs, and parameter outputs would clarify the presentation and may in turn resolve any concerns about the strength of evidence.

Additional Major Comments/Questions

1) The data in reference [1] derive from RNA-seq. The authors mention the use of UMIs to correct PCR amplification, but how do they deconvolute clonal expansion from transcript abundance? This is a major issue when using RNA-seq data for which cells were not barcoded.

2) It is not clear what data is being used for which procedures. When is real data being used versus when is synthetic data being used?

a) Page 3: "Fitting is performed with an expectation-maximization algorithm which finds maximum-likelihood estimates of the prevalence ˆρ and mean intrafamily distance ˆµ for each VJl class." Was the fit with synthetic data generated using soNNia, as indicated for P0(x)? Or real data from [1], as indicated in Figure 1A?

b) Page 3: What data were used for Figures 1B-1I? Data from reference [1] are used in A, and the dashed lines in J are from synthetic data. What about the rest? All from [1]?

c) Similar questions persist throughout.

3) The authors need to be clear when they are drawing conclusions from real data versus from synthetic data. And in general, real data should be used as much as possible, since synthetic data is limited due to the generation process being based only on what we think we already know. On the other hand, I understand that for real data, we don't ever have gold standard labels, so a careful balance of the two is needed.

a) Page 3: "P0(x) = P0(x|l) is computed for each length l by generating a large number of unrelated, same-length sequences with the soNNia model of recombination and selection [19], and calculating the distribution of their pairwise distances." How does this compare to using real data? What if you, e.g., take all singletons in a real dataset? Wouldn't that also approximate P0(x)? Figure 2G shows the peak of the distribution of x at 0.5, half of all nucleotide positions differ; assuming this is from simulated data, is this what you would get with real data? Or vice versa.

b) Page 3: "The results of the fit show that ˆµ varies little between VJl classes, around ˆµ ∼ 4%." Is this based on real or synthetic data? If synthetic data, does this just depend on how you wrote the generation process? This may be related to the questions below regarding how mutations were identified and characterized and then applied in the data generation process.

c) Page 3: "In contrast, the prevalence ˆρ varies widely across classes". Again, real or synthetic data? If synthetic, doesn't this depend on the generation process? If real data, is there any association with l?

d) Similar questions persist throughout.

4) Page 4: "it is expected to fail when the prevalence and the CDR3 length are both low." A single value rho can encompass a large number of small clones or a small number of large clones, but rho seems to be treated in the model as a repertoire level rho, the proportion of all pairs that are related. How does this impact the interpretation/expectation of the approach's performance across different tissue types or B cell subsets that may be expected to have very different clonal distributions that are averaged out to a single rho in the model?

5) Page 4: "We first estimated the distribution of clonal family sizes from the data of [1] by applying the CDR3-based clustering method with adaptive threshold.… to VJl classes for which.… the predicted sensitivity was >90%." Does this mean VJl classes with e.g., l>30? (inferring from Figure 2I) Or what does this mean? Could focusing only on these clusters bias the estimated distribution of clone sizes?

6) Page 4: "For each lineage we draw a random progenitor using soNNia". I assume these are lineages for the synthetic data, but it isn't clear from the text that you aren't sampling from the lineages created for [1]. Similarly, on page 10: "To generate synthetic data we make use of the lineages identified in the high-sensitivity and high-precision regime of CDR3-based inference (Figure 3F), we denote the set of these lineages by L." That means you use the lineages identified in the data from [1]? And should the figure reference be 3E?

7) Page 4: "Mutations are then randomly drawn on each sequence of the lineage in a way that preserves the mutation sharing patterns observed in families of comparable size from the partitioned data". What partitioned data? Data from reference [1]? And how were the mutation patterns characterized so they could be replicated? From page 10: "We then identify all unique mutations in the true lineage and for each mutation denote the labels of members of the lineage that carry it." How are "all unique mutations in the true lineage" identified at scale? And how is the "true lineage" identified? Using the approach being developed, I think, so this all seems a bit circular.…

8) Page 11: "We compute the expected distributions of the CDR3 Hamming distance n, and the number of shared mutations n0, under a uniform mutation rate assumption. In other words, we assume that the probability that a given position was mutated, given a mutation happened somewhere in a sequence of length L, equals L^{-1}." I think it is well-known that this is not the case. What are the implications for the evaluation of model performance?

*Reviewer #2 (Recommendations for the authors):*

Please add line numbers: they make reviewing much easier.

Overall it would be helpful to understand how "precision" is used. "High-precision" comes up a lot, including in the HILARy acronym, and it seems that sometimes it's used as a synonym for accuracy. Or is it literally "precision" in the technical sense, i.e. contrasted to recall/sensitivity?

The mathematical exposition is nicely laid out. However, the choices of symbols can make it difficult to follow, and the reader has to keep a lot of arbitrarily-chosen notation in their head to follow and read figures. A notation table would help, as would more self-explanatory naming choices for key variables like x, y, n, and n_0. For example, it would make things easier to read to use something like P_F rather than P_0, and P_T rather than P_1. 0 and 1 subscripts are used in a different context when used as a subscript in equation (4), which makes this equation difficult to parse: sometimes 1 means the number of mutations in sequence 1, and sometimes it means the same-lineage assumption.

Results

A

- "which consists in" suggest "which consists of"

- prevalence is a key definition, and right now it's tacked onto the end of a sentence. Suggest making it a stand-alone definition for clarity.

- "share the same V and J gene usage" suggest "share the same V and J genes"

- "The signature of the VDJ rearrangement is largely encoded by the CDR3 alone" this statement seems over-strong-- the parts of V genes within the CDR3 are frequently identical.

- Productive sequences are often defined as not only cdr3 length divisible by 3, but also V and J in frame (indels could change that).

- "same-length sequences with the soNNia model": do these include mutation?

- If I understand correctly the Poisson distribution models the distribution of distances within a clonal family, and so mu is a function of the distribution of the number of somatic hypermutations. If this is correct, it would be helpful to include this interpretation.

- "μ varies little between VJl classes, around ˆμ ∼ 4%": suggest using $\simeq$ for approximate as done below

- "that the positive mode P1(x) of the distribution varies little," suggest "… the mode of the positive distribution P_1(x)"

B

- Suggest mentioning that the results here depend on the level of SHM.

C

- "verify that these performance predictions hold in real inference tasks": I would recommend "realistic" rather than "real", the latter of which sounds like you mean real data.

D

- Below (4), _1 should be _1.

- Suggest using more precise language to define n rather than "divergence", describing that this is about pairs of sequences.

- "computationally expansive" should be "computationally expensive"

Figure 3

- What causes the three bent stripes in C?

- It would be really interesting to know how this plot changes with SHM: in C, higher SHM will result in more divergence among positive CDR3s (more density at higher x).

Perhaps this will be offset by more shared mutation (i.e. maybe new high-SHM points will stay above your straight line x'-y=t'), but it would be really great to find out.

- I don't think it's clear either here or in the text what is actually done with the high-precision and high-sensitivity partitions.

The bottom right of p5 (in Results) partially explains a use for the high-sensitivity one (but not high-precision), and the D section of Methods describes how to calculate them (but not what is done with them).

F

- How was dN/dS calculated? Traditionally this has been done in a maximum-likelihood setting to calculate rates, but is this more of a counting approach?

Methods

A

- "remplates" – "templates"

- I can't seem to find in reference [32] any measurement of how incorrect V and J assignments affect initial VJl partitioning, could you clarify which figure this is from?

C

- "In case of l class": is there a typo here? I'm not sure what this means.

- two equal signs in (10)

D

- The sentences following "This crude method suffers from inaccuracy as it loses precision in the case of highly-mutated sequences and junctions of short length" describe how your prefix tree implementation speeds up this crude approach, but not how it improves on the accuracy.

- "standard methods [use a] fixed threshold on Hamming distance divergence"

The standard methods use a threshold on the fraction of differing CDR3 bases, i.e. the threshold on divergence depends on (is proportional to) CDR3 length.

- Could you make more explicit how the increased speed allows for an "adaptive threshold"?

The text after equation (17) seems to suggest that several reruns are involved, but it could be made more clear.

- I don't understand the purpose of constructing these various partitions.

Ostensibly they seem to be to help calculate a threshold, but the section seems to finish having just constructed the partitions, without telling me either how they've helped you arrive at a threshold, or why that threshold is so much better than what other methods have used.

F

(24) I think there should also be an approximate equals sign in the first line in this equation because we are substituting in the approximate value of p

(25) Suggest pointing out that this follows because we are approximating the distribution as Poisson

- suggest "analogously" rather than "analogically" https://english.stackexchange.com/questions/112149/analogous-vs-analogical

- "is chosen based on the prevalence, analogically to adaptive threshold"

I think you're missing a prime after the t in the inequality in this sentence

- "and further assume n0 ∼ n1n2 L to compute the new null distribution"

I think that this tilde means "is distributed as" because n_1 and n_2 are random variables. A little clarification would help here.

*Reviewer #3 (Recommendations for the authors):*

In addition to the recommendations in the public review, we had the following more questions and recommendations:

P. 2 "Low prevalence is usually due to a high frequency of singletons" Can the authors demonstrate this, or provide a citation?

P. 2 "In addition, we restrict our analysis to CDR3 lengths between 15 and 105: shorter or longer junctions have comparable frequencies to nonproductive junctions, suggesting that they are nonfunctional." This should be shown somewhere or given a citation.

P. 3 It is not clear how mu can vary so little but rho can vary so much

P. 3 use of rho_hat and p_hat is confusing

It would be helpful to see more descriptive statistics of the simulated data, such as the level of somatic hypermutation.

Figure 1 B. From the coloring of the tree/cluster, it looks like two sequences are left out of the tree.

P. 9 Throughout the paper, does "distance" always refer to Hamming distance?

P. 11 Uniform mutation probabilities across BCRs is not biologically realistic, given SHM hotspots as well as CDR and FWR regions. We understand this was a simplifying assumption, but it should be discussed.

P. 11 Potential typo: x' – y <= t. Should this be x' – y <= t'?

[Editors' note: further revisions were suggested prior to acceptance, as described below.]

Thank you for resubmitting your work entitled "Combining mutation and recombination statistics to infer clonal families in antibody repertoires" for further consideration by *eLife*. Your revised article has been evaluated by Miles Davenport (Senior Editor) and a Reviewing Editor.

The manuscript has been improved but there are some remaining issues that need to be addressed, as outlined below:

*Reviewer 2*

We thank the authors for their careful consideration of our comments, and for the many changes that they have implemented. Our main concerns have been addressed, although we have several minor comments.

The authors have made a huge improvement to the documentation and useability. A minor point, however, is that the result file seems to mostly be in AIRR format, except that the AIRR-standard 'clone_id' column for clonal family information seems to be either absent or copied from the input file, while an undocumented 'family' column appears to hold the inferred family information. This is confusing.

It would be wonderful if the authors chose to go for AIRR software certification as described in https://docs.airr-community.org/en/latest/swtools/airr_swtools_standard.html

We agree that HILARy makes a very real performance improvement in distinguishing similar, but unrelated, families, which is a substantial advance. However, in digging through things a bit more, the paper doesn't seem to clearly communicate how frequently settings arise in which HILARy is substantially different than existing software.

First, the performance plots in Figure 4 only show performance for half of CDR3 lengths (15-45, compared to 15-80 shown in the CDR3 distribution in Figure 6a).

Furthermore, the CDR3 lengths at which other methods perform significantly worse than HILARy (lengths 15-24) constitute less than one percent of a typical repertoire (see for instance Figure 6a). While it's true that it can make sense to emphasize challenging regions of parameter space, it would seem reasonable to clarify that this is what is being presented, as well as showing performance on typical repertoires in order to show the relevant context.

The simulation samples also seem to use a restricted set of J genes: 80% of sequences are from a single J allele, whereas repertoires more typically use around four J alleles with prevalences of perhaps 10% to 50%. This also has the effect of inflating the number of very similar but unrelated families. This is an example of the risks of focusing on a single data sample for validation.

We also have one small clarification in response to:

"We agree that testing the method on a differently generated dataset is a useful check. We should point out, however, that our synthetic dataset is not as biased as it may seem. In particular, it is based on trees from VJl classes that we predicted are very easy to cluster, which means that they are truly faithful to the data, and not dependent on the particular algorithm used to infer them. The big advantage over this synthetic dataset over others is that it recapitulates the power law statistics of clone size distribution, as well as the diversity of mutation rates. To us, it still represents a more useful benchmark than synthetic datasets generated by population genetics models, which miss most of this very broad variability."

We just want to clarify that our concern was not that the synthetic sample was biased, but rather that it is risky to rely on any single sample, whether data or simulation, to form the basis of a robust inference method. Any single sample represents only a single set of possible parameter values, whereas we generally want methods to work well on real data samples that exist at a huge variety of different parameter values.

Also, while deciding when to use synthetic vs real data is a challenging problem, and we don't in general find fault with the authors' choices about this, we do want to point out that the authors' suggestion here that simulation methods cannot mimic data-like clone size distributions or mutation variability (i.e. tree shape) is incorrect. There exist simulation methods that let the user configure these parameters arbitrarily (including using direct inference from data).

*Reviewer 3*

The authors have addressed the major points I brought up in my review, and the revision is definitely improved. With respect to their new benchmarking on single linkage hierarchical clustering (Figure 4, supplement 2), it's not clear to me why they used their own implementation rather than one of the existing tools, or why the thresholds of 0.76, 0.82, and 0.88 were chosen. However, I think these issues could be addressed with text edits. Personally, I'm trying to reconcile the results of this study with a recent and more comprehensive clonal benchmarking study which showed that single linkage clustering worked quite well: https://bmcimmunol.biomedcentral.com/articles/10.1186/s12865-024-00600-8. May be worth adding some discussion about.

I agree with Reviewer 2 above points about consistency with AIRR format and that (1) the method seems to perform well in a CDR3 space which represents a small fraction of typical repertoire space, so it's not terribly clear how much it improves overall performance, and (2) it's risky to rely on effectively a single dataset to base these conclusions on.

---

## [Author Response]

Essential revisions:1) Provide clarification regarding how different data sets (synthetic versus real) were used for different steps during algorithm development and validation, perhaps by including a flow chart of inputs/outputs/dependencies.

We have added clarifications throughout, in the main text as well as in the caption of Figure 2, to clarify what is data and what is model. We did not include a flow chart, because it’s hard to think of ways to visualize how data is being used, since it comes in two steps: in the first step, the algorithm devises what thresholds to use using statistics from the data; in the second step, it actually performs the clustering on the data. However, we have heavily edited the text to better explain the procedure. We now summarize the workflows in a series of steps for both methods (HILARy-CDR3 and -full) at the end of their respective sections. We have also added a table of notations to make it easier to follow the steps of the approach.

2) Conduct an evaluation that is not circular, such as using additional simulated data sets beyond the one based on the same data used to develop the method.

We have added a new row of benchmark results to Figure 4 to show the results of the benchmark on an independent, model-generated synthetic dataset from the group that designed the partis method. Our method still performs competitively, on par with partis—which was developed and tested on that dataset—and better than other methods.

In addition, we have used that dataset to benchmark a new version of HILARy that also uses the light chain. We compare its performance to other methods on the same synthetic dataset from the partis team in new Figure 5 (in the same fashion as Figure 4), and in Figure 4—figure supplement 1 to show the dependence as a function of mutation rate.

3) Provide details about the versions and settings of the competitor programs included in the benchmarking, as well as other general information such as: what type of machine was used? Was the use of multiple threads enabled for all programs? When multiple methods are available within a single program (e.g., scoper), indicate which were used and justify the choice. Etc.

We have added all these details in the Methods section, as well as in the github, where the scripts used are also shared.

4) Make the synthetic data publicly available as well as the scripts used for the benchmarking analysis.

We have made the synthetic data publicly available on zenodo (https://zenodo.org/ records/10676371) and added the scripts used for benchmarking to HILARy’s github.

5) Define "precision" prominently and use it in that sense consistently throughout the manuscript.

By “precision" we mean the statistical measure also called positive predictive value. We added this clarification the first time the word is used in the introduction. This comes in addition to its formal definition in the main text: “precision pi(t), defined as a proportion of true positives among all pairs classified as positive (Figure 2E)”, and illustration by Figure 2E. We have also added a table of notations where precision is clearly defined (second definition).

Reviewer #1 (Recommendations for the authors):Regarding point 1 in the public review, examples of more specific questions along these lines are included below. A flow chart figure showing the steps, data inputs, and parameter outputs would clarify the presentation and may in turn resolve any concerns about the strength of evidence.Additional Major Comments/Questions1) The data in reference [1] derive from RNA-seq. The authors mention the use of UMIs to correct PCR amplification, but how do they deconvolute clonal expansion from transcript abundance? This is a major issue when using RNA-seq data for which cells were not barcoded.

We consider unique sequences and therefore transcript abundance does not influence our downstream analysis. The sizes of clonal families inferred from data [1] are reported in terms of the number of unique sequences.

We have clarified this point in the methods.

2) It is not clear what data is being used for which procedures. When is real data being used versus when is synthetic data being used?

We have tried to clarify when real data, models, and synthetic data were used. The discussion is subtle, because the null models used to estimate the threshold are themselves fit to data.

a) Page 3: "Fitting is performed with an expectation-maximization algorithm which finds maximum-likelihood estimates of the prevalence ˆρ and mean intrafamily distance ˆµ for each VJl class." Was the fit with synthetic data generated using soNNia, as indicated for P0(x)? Or real data from [1], as indicated in Figure 1A?

The fit is of the data distribution from [1]. The model-generated P0 (now called P_F) is part of the mixture model along with and P1 (now called P_T).

We have clarified the fitting procedure and what’s data and model.

b) Page 3: What data were used for Figures 1B-1I? Data from reference [1] are used in A, and the dashed lines in J are from synthetic data. What about the rest? All from [1]?

It is not easy to answer this question because some figures are model prediction, but still based on adjustable parameters that were fit to each VJl class from the data.

We have tried to clarify the source of the plotted data both in the main text and in the caption. We have added a ref to the donor for Figure 2B, and explained that Figure 2C-F are just illustrative. We have replaced “null” by “model” in Figure 2G. We have explained that Figure 2H-J are estimated a priori from the model for chosen values of mu and rho, which we now say explicitly

c) Similar questions persist throughout.

We hope that the clarifications above will dissipate further ambiguities.

3) The authors need to be clear when they are drawing conclusions from real data versus from synthetic data. And in general, real data should be used as much as possible, since synthetic data is limited due to the generation process being based only on what we think we already know. On the other hand, I understand that for real data, we don't ever have gold standard labels, so a careful balance of the two is needed.

We prioritized simplicity and robustness. The null models were chosen to be simple and/or independently inferred from large datasets (soNNia model). The adjustable (data-driven) part of the model is reduced to the two parameters mu and rho, which are re-learned for each VJl class.

We have made it clearer what is fitted and with what parameters, and have also added a summary of the flow at the end of the 2nd and 4th sections of the Results.

a) Page 3: "P0(x) = P0(x|l) is computed for each length l by generating a large number of unrelated, same-length sequences with the soNNia model of recombination and selection [19], and calculating the distribution of their pairwise distances." How does this compare to using real data? What if you, e.g., take all singletons in a real dataset? Wouldn't that also approximate P0(x)? Figure 2G shows the peak of the distribution of x at 0.5, half of all nucleotide positions differ; assuming this is from simulated data, is this what you would get with real data? Or vice versa.

We used soNNia because we found that it was very good at estimating the distribution of sequences (as demonstrated in Ref. 19). Singletons could be biased relative to other sequences. In the method we explore the possibility of computing the distance of sequences belonging to the real data and generated by soNNia as an alternative null model called P’_F. Further tests of the model against real data are beyond the scope of this paper, where the adequacy of our approximations is ultimately measured in terms of performance.

b) Page 3: "The results of the fit show that ˆµ varies little between VJl classes, around ˆµ ∼ 4%." Is this based on real or synthetic data? If synthetic data, does this just depend on how you wrote the generation process? This may be related to the questions below regarding how mutations were identified and characterized and then applied in the data generation process.

These are real data estimates, or rather from the fit to the data.

We have now given the data source and given explanations of the fit, to make it clearer that this number is inferred from data.

c) Page 3: "In contrast, the prevalence ˆρ varies widely across classes". Again, real or synthetic data? If synthetic, doesn't this depend on the generation process? If real data, is there any association with l?

See above. We did not see any association of rho with length.

d) Similar questions persist throughout.

We hope that the clarifications above will dissipate further ambiguities.

4) Page 4: "it is expected to fail when the prevalence and the CDR3 length are both low." A single value rho can encompass a large number of small clones or a small number of large clones, but rho seems to be treated in the model as a repertoire level rho, the proportion of all pairs that are related. How does this impact the interpretation/expectation of the approach's performance across different tissue types or B cell subsets that may be expected to have very different clonal distributions that are averaged out to a single rho in the model?

rho is inferred for each VLl class, so is not a repertoire-wide property. We have clarified that rho was inferred for each VJl class.

Repertoire heterogeneity is an interesting question but this (or other) inference procedure does not use subset or tissue information, as that information is often not available. Using phenotypic information to inform clonal inference procedure is a nice idea and could be the object of future work. The fact that most dataset are mixtures of phenotypes, specificities, and immune histories, each with their own clone size distribution and possibly sequence biases, is an inherent difficulty that all lineage clustering algorithms face.

5) Page 4: "We first estimated the distribution of clonal family sizes from the data of [1] by applying the CDR3-based clustering method with adaptive threshold.… to VJl classes for which.… the predicted sensitivity was >90%." Does this mean VJl classes with e.g., l>30? (inferring from Figure 2I) Or what does this mean? Could focusing only on these clusters bias the estimated distribution of clone sizes?

Operationally, what this means is strictly defined by a predicted sensitivity of >90%, as written in the text. As the reviewer correctly intuits, this correlates with large l, but it also depends on the prevalence inferred for that class (which is fixed in Figure 2I).

We hope that the earlier clarification about how figure 2 is exactly plotted will have dissipated this confusion.

6) Page 4: "For each lineage we draw a random progenitor using soNNia". I assume these are lineages for the synthetic data, but it isn't clear from the text that you aren't sampling from the lineages created for [1]. Similarly, on page 10: "To generate synthetic data we make use of the lineages identified in the high-sensitivity and high-precision regime of CDR3-based inference (Figure 3F), we denote the set of these lineages by L." That means you use the lineages identified in the data from [1]? And should the figure reference be 3E?

We apologize if this procedure was not clearly explained. The only part of the data we used to generate our synthetic data is the lineage size distribution, and the statistics of tree topologies. 3F is the correct reference, as it shows in yellow the sets of prevalences and VJl classes for which sensitivity is high enough for our inclusion criterion.

We have re-written the explanation in the main text, and have clarified in the Methods that we only use the statistics of tree topologies. We have also clarified that Figure 3F referred to the region above the black line.

7) Page 4: "Mutations are then randomly drawn on each sequence of the lineage in a way that preserves the mutation sharing patterns observed in families of comparable size from the partitioned data". What partitioned data? Data from reference [1]?

We have clarified that the lineages L \mathcal{L} were inferred from data.

And how were the mutation patterns characterized so they could be replicated?

This is explained in the part about the labels of lineage nodes that carry a particular mutation.

We have added “To create a lineage with the same mutation patterns as the real data…” to explain that this is what it is about.

From page 10: "We then identify all unique mutations in the true lineage and for each mutation denote the labels of members of the lineage that carry it." How are "all unique mutations in the true lineage" identified at scale?

They are identified using sequence alignment and tree reconstruction methods already described in Spisak et al. (Ref. 33).

We have rephrased the sentence and added the reference.

And how is the "true lineage" identified? Using the approach being developed, I think, so this all seems a bit circular.…

The true lineages were identified using the “lineages identified in the high- sensitivity and high- precision regime of CDR3-based in- ference from the data (yellow region above the black line in Figure 3F)”, as now better explained in the text.

The argument is not really circular, because those lineages are for VJl classes for which the inference was highly reliable, because of the high discriminability of positive and negative pairs. Besides, to create the synthetic data the sequences are actually “erased” from the trees, and replaced by entirely new sequences and mutations but respecting the tree topology (or said differently genealogical structure of mutations), including some with much more difficult VJl class.

We have added an explanation in the main text of why the re-use of real data lineages inferred by HILARy doesn’t bias the procedure, since it’s done on a subset of lineages within VJl classes that are easy to infer.

8) Page 11: "We compute the expected distributions of the CDR3 Hamming distance n, and the number of shared mutations n0, under a uniform mutation rate assumption. In other words, we assume that the probability that a given position was mutated, given a mutation happened somewhere in a sequence of length L, equals L^{-1}." I think it is well-known that this is not the case. What are the implications for the evaluation of model performance?

This is a simplifying assumption that makes the computation easier. Accounting for the mutatability heterogeneity could only improve the performance of the method.

We have clarified that this assumption is not meant to be realistic, but computational practical.

Reviewer #2 (Recommendations for the authors):Please add line numbers: they make reviewing much easier.

We have added line numbers.

Overall it would be helpful to understand how "precision" is used. "High-precision" comes up a lot, including in the HILARy acronym, and it seems that sometimes it's used as a synonym for accuracy. Or is it literally "precision" in the technical sense, i.e. contrasted to recall/sensitivity?

Precision always refers to the total of true positives over the total of positives TP/(TP+FP), as defined the first time the word appears in the results: “defined as a proportion of true positives among all pairs classified as positive”, and illustrated by Figure 2E.

We noticed the word precision is also used in the introduction. We added the equivalent term “positive predicted value” there to make clear from the beginning that we refer to a statistical quantity with a standard definition. We have also added a table of definitions of notations where precision is defined again.

The mathematical exposition is nicely laid out. However, the choices of symbols can make it difficult to follow, and the reader has to keep a lot of arbitrarily-chosen notation in their head to follow and read figures. A notation table would help, as would more self-explanatory naming choices for key variables like x, y, n, and n_0. For example, it would make things easier to read to use something like P_F rather than P_0, and P_T rather than P_1. 0 and 1 subscripts are used in a different context when used as a subscript in equation (4), which makes this equation difficult to parse: sometimes 1 means the number of mutations in sequence 1, and sometimes it means the same-lineage assumption.

We have changed P_1 and P_0 to P_F and P_T.

We have added a table (Table I) with a summary of notations.

Results

A

- "which consists in" suggest "which consists of"Corrected.prevalence is a key definition, and right now it's tacked onto the end of a sentence. Suggest making it a stand-alone definition for clarity.

We have now highlighted that definition of prevalence in its own sentence.

Corrected."The signature of the VDJ rearrangement is largely encoded by the CDR3 alone" this statement seems over-strong-- the parts of V genes within the CDR3 are frequently identical.

We have modified the sentence.

The statement is that the CDR3 is often enough to distinguish distinct recombination events, and does not imply that the entire CDR3 is variable.

- Productive sequences are often defined as not only cdr3 length divisible by 3, but also V and J in frame (indels could change that).

The reviewer is correct but these sequences, since they entered affinity maturation to receive an indel, were initially productive, and thus belong to what we define as productive lineages, even if they are themselves non productive.

We have clarified the text that the criterion of productivity is really about the whole lineage, and thus about the ancestral (naive) sequence (which cannot have indels since it’s not mutated yet).

- "same-length sequences with the soNNia model": do these include mutation?

They do not. However junctional diversity is hard to distinguish for further mutational diversification, and in practice soNNIa is a good model.

We have clarified this point in the methods.

- If I understand correctly the Poisson distribution models the distribution of distances within a clonal family, and so mu is a function of the distribution of the number of somatic hypermutations. If this is correct, it would be helpful to include this interpretation.

We have added an explanation.

- "μ varies little between VJl classes, around ˆμ ∼ 4%": suggest using $\simeq$ for approximate as done below

Corrected.

- "that the positive mode P1(x) of the distribution varies little," suggest "… the mode of the positive distribution P_1(x)"

We have rephrased this sentence after realizing that “mode” is ambiguous, as it may refer to the value of the peak, or the peak itself.

B- Suggest mentioning that the results here depend on the level of SHM.

We know interpret mu as a proxy for the SHM. We believe the text clearly explains that all inferences depend on mu, however mu is fairly constant across VJl classes, so that the results actually do not depend much on it.

At the same time, the reviewer is right that performance may in general depend on mu, which here is fixed in the dataset. We have added a benchmark that allows us to study this dependence. The results are reported in new the Figure 4—figure supplement 1.

C- "verify that these performance predictions hold in real inference tasks": I would recommend "realistic" rather than "real", the latter of which sounds like you mean real data.

Corrected.

D- Below (4), _1 should be _1.

Corrected.

- Suggest using more precise language to define n rather than "divergence", describing that this is about pairs of sequences.

We replaced it with “distance”.

Figure 3What causes the three bent stripes in C?

They correspond to different (discrete) values of the denominator in y, depending on n_0 (n_0=1,2,3,…).

- It would be really interesting to know how this plot changes with SHM: in C, higher SHM will result in more divergence among positive CDR3s (more density at higher x).Perhaps this will be offset by more shared mutation (i.e. maybe new high-SHM points will stay above your straight line x'-y=t'), but it would be really great to find out.

We have added new Figure 4—figure supplement 1 to study just that, based on the dataset from Ralph et al. 2022 (used to test partis) where the mutation rate is controllable.

- I don't think it's clear either here or in the text what is actually done with the high-precision and high-sensitivity partitions.The bottom right of p5 (in Results) partially explains a use for the high-sensitivity one (but not high-precision), and the D section of Methods describes how to calculate them (but not what is done with them).

The general idea is to reduce computations, in particular of pairwise comparisons, which scale with the square of the number of sequences. The high-sensitivity and high-precision partitions are nested partitions. One is too coarse, and the other is too fine. The full HILARy method is then used to merge fine clusters within the coarse ones.

We have changed the text in the main text as well as in the methods to explain this better.

F- How was dN/dS calculated? Traditionally this has been done in a maximum-likelihood setting to calculate rates, but is this more of a counting approach?

The reviewer is correct. We used the simplest approach, since we were not after estimating the quantity and interpreting its absolute value but only after assessing variability of the distribution over families with a given CDR3 length.

Methods

A

- "remplates" – "templates"Corrected. - I can't seem to find in reference [32] any measurement of how incorrect V and J assignments affect initial VJl partitioning, could you clarify which figure this is from?

The statement is not directly whether they affect partitioning but that these events are rare. We have clarified “such indel events”.

C- "In case of l class": is there a typo here? I'm not sure what this means.

We have rephrased the sentence.

- two equal signs in (10)

Corrected.

D- The sentences following "This crude method suffers from inaccuracy as it loses precision in the case of highly-mutated sequences and junctions of short length" describe how your prefix tree implementation speeds up this crude approach, but not how it improves on the accuracy.

You’re right that these are separate points. We have removed the “However”.

- "standard methods [use a] fixed threshold on Hamming distance divergence"The standard methods use a threshold on the fraction of differing CDR3 bases, i.e. the threshold on divergence depends on (is proportional to) CDR3 length.

Corrected.

- Could you make more explicit how the increased speed allows for an "adaptive threshold"?The text after equation (17) seems to suggest that several reruns are involved, but it could be made more clear.

Apologies for the confusion but the adaptive threshold and prefix trees are separate innovations. The prefix tree improves speed, the adaptive threshold improves accuracy.

We have clarified.

We have rewritten the part following eq. 17.

The clustering is run only twice to obtain the coarse (high sensitivity) and fine (high precision) partitions.

- I don't understand the purpose of constructing these various partitions.Ostensibly they seem to be to help calculate a threshold, but the section seems to finish having just constructed the partitions, without telling me either how they've helped you arrive at a threshold, or why that threshold is so much better than what other methods have used.

This is just the first step before applying the mutation-based method, and its purpose is to reduce computations.

We now explain this, and added a paragraph in the Methods F and a supplementary figure to explain this better. This is also now better explained in the main text in section D.

F(24) I think there should also be an approximate equals sign in the first line in this equation because we are substituting in the approximate value of p

Corrected

(25) Suggest pointing out that this follows because we are approximating the distribution as Poisson

It actually follows from the fact that n1 and n2 are small relative to L. The consequence is that it becomes a Poisson.

We have clarified.

- suggest "analogously" rather than "analogically" https://english.stackexchange.com/questions/112149/analogous-vs-analogical

This paragraph has been edited and that word doesn’t appear anymore.

- "is chosen based on the prevalence, analogically to adaptive threshold"I think you're missing a prime after the t in the inequality in this sentence

Solved. This paragraph was re-written.

- "and further assume n0 ∼ n1n2 L to compute the new null distribution"I think that this tilde means "is distributed as" because n_1 and n_2 are random variables. A little clarification would help here.

Solved. This paragraph was re-written.

Reviewer #3 (Recommendations for the authors):In addition to the recommendations in the public review, we had the following more questions and recommendations:P. 2 "Low prevalence is usually due to a high frequency of singletons" Can the authors demonstrate this, or provide a citation?

We removed that sentence.

P. 2 "In addition, we restrict our analysis to CDR3 lengths between 15 and 105: shorter or longer junctions have comparable frequencies to nonproductive junctions, suggesting that they are nonfunctional." This should be shown somewhere or given a citation.

We have rewritten this part to better motivate this choice.

P. 3 It is not clear how mu can vary so little but rho can vary so much

It is not completely clear why that is the case, but this is an empirical observation. An interpretation is that since mu is linked to mean mutation rate, this result suggests that it is stable across classes. rho is linked to the clone size distribution, and is very sensitive to the exponent of its power law. This could explain why it is very variable. The method does not make assumption about this variability infers it for each class on each dataset.

P. 3 use of rho_hat and p_hat is confusingIt would be helpful to see more descriptive statistics of the simulated data, such as the level of somatic hypermutation.

The level of hypermutation in our synthetic data is the same as in the actual data, since it mimics its mutational structure, with around mu=4% divergence between related sequences.

We now give this mean divergence in the main text.

The second synthetic dataset (from the partis paper) on which we benchmark the methods allows us to control the SHM rate. We have added a Figure 4—figure supplement 1 to show the dependence of performance on that mutation rate.

Figure 1 B. From the coloring of the tree/cluster, it looks like two sequences are left out of the tree.

We have changed the figure to make it more consistent. The root of the tree is unobserved and represented in white.

P. 9 Throughout the paper, does "distance" always refer to Hamming distance?

Yes, distance is always the Hamming distance n. We have added a table with notations, in which we clearly n as the Hamming distance and x=n/l as its normalized version.

P. 11 Uniform mutation probabilities across BCRs is not biologically realistic, given SHM hotspots as well as CDR and FWR regions. We understand this was a simplifying assumption, but it should be discussed.

We have added a discussion of this approximation in the methods, with a reference.

P. 11 Potential typo: x' – y <= t. Should this be x' – y <= t'?

Yes. We have changed that paragraph and the problem is now solved.

[Editors' note: further revisions were suggested prior to acceptance, as described below.]

The manuscript has been improved but there are some remaining issues that need to be addressed, as outlined below:Reviewer 2We thank the authors for their careful consideration of our comments, and for the many changes that they have implemented. Our main concerns have been addressed, although we have several minor comments.The authors have made a huge improvement to the documentation and useability. A minor point, however, is that the result file seems to mostly be in AIRR format, except that the AIRR-standard 'clone_id' column for clonal family information seems to be either absent or copied from the input file, while an undocumented 'family' column appears to hold the inferred family information. This is confusing.It would be wonderful if the authors chose to go for AIRR software certification as described in https://docs.airr-community.org/en/latest/swtools/airr_swtools_standard.html

We have changed the ‘family’ column label to ‘clone_id’.

We agree that HILARy makes a very real performance improvement in distinguishing similar, but unrelated, families, which is a substantial advance. However, in digging through things a bit more, the paper doesn't seem to clearly communicate how frequently settings arise in which HILARy is substantially different than existing software.First, the performance plots in Figure 4 only show performance for half of CDR3 lengths (15-45, compared to 15-80 shown in the CDR3 distribution in Figure 6a).

We agree that we should have clearly stated the range of parameters for which HILARy distinguishes itself from other software.

We have now clarified that HILARy is substantially different when the CDR3 is short or the hypermutation rate is high. We have also retitled the corresponding result section ‘Benchmark of the methods on heavy chain datasets’ and included the performance relative to mutation rates for heavy chains in the main text (see new Figure 5).

Additionally, we would like to highlight that (1) HILARy is the only software that scales with the number of sequences in the largest VJl class (see Figure 4A), and (2) when using paired heavy and light chains, HILARy demonstrates a high degree of comparability to Partis software across all mutation rates, as evidenced by tests conducted on a simulated dataset specifically created for benchmarking Partis (see Figure 6).

Furthermore, the CDR3 lengths at which other methods perform significantly worse than HILARy (lengths 15-24) constitute less than one percent of a typical repertoire (see for instance Figure 6a). While it's true that it can make sense to emphasize challenging regions of parameter space, it would seem reasonable to clarify that this is what is being presented, as well as showing performance on typical repertoires in order to show the relevant context.

We do not focus on CDR lengths greater than 45 because in this range reconstructing clonal families is expected to be easier, and indeed all methods tested here performed well.

We have added a clarification in the benchmark section and extended the range of CDR3 lengths to 15-60 on our synthetic data in our benchmark against hierarchical clustering in the supplementary information (Figure S10) to provide evidence that longer CDRs do not affect HILARy’s performance.

In addition, we should emphasize that, for many analysis, it is crucial to guarantee consistent sensitivity and precision. Conversely, using the same threshold for all CDR3 lengths could confound analysis of clonal selection and dynamics. For example, in a study by Horns et al. 2019, the lineages detected as "persistent" and "vaccine-responsive" have very different CDR3 lengths (see Author response image 1 where we used the lineage assignment and labeling directly from Horns et al.; the original paper didn't show it). The persistent lineages, which are of biological interest, include short CDR3 and although they do not make up a significant portion of the repertoire, identifying them correctly is important for the biological conclusions.

**Author response image 1. sa2fig1:** 

We have reported performance on a typical repertoire in the Results and added a paragraph in the Discussion section to clarify that simpler methods can be used depending on the repertoire’s statistics.

The simulation samples also seem to use a restricted set of J genes: 80% of sequences are from a single J allele, whereas repertoires more typically use around four J alleles with prevalences of perhaps 10% to 50%. This also has the effect of inflating the number of very similar but unrelated families. This is an example of the risks of focusing on a single data sample for validation.

It is true that the IGJH4 gene is largely represented in the simulation sample, but this is due to the fact that it is also largely represented in the data of [1]. Looking at the frequency of this gene across CDR3 lengths, we find our simulation sample to be comparable to the real data. See Author response image 2.

[1]Briney B, Inderbitzin A, Joyce C, Burton DR (2019) Commonality despite exceptional diversity in the baseline human antibody repertoire. Nature 566:393–397.

We also have one small clarification in response to:"We agree that testing the method on a differently generated dataset is a useful check. We should point out, however, that our synthetic dataset is not as biased as it may seem. In particular, it is based on trees from VJl classes that we predicted are very easy to cluster, which means that they are truly faithful to the data, and not dependent on the particular algorithm used to infer them. The big advantage over this synthetic dataset over others is that it recapitulates the power law statistics of clone size distribution, as well as the diversity of mutation rates. To us, it still represents a more useful benchmark than synthetic datasets generated by population genetics models, which miss most of this very broad variability."We just want to clarify that our concern was not that the synthetic sample was biased, but rather that it is risky to rely on any single sample, whether data or simulation, to form the basis of a robust inference method. Any single sample represents only a single set of possible parameter values, whereas we generally want methods to work well on real data samples that exist at a huge variety of different parameter values.Also, while deciding when to use synthetic vs real data is a challenging problem, and we don't in general find fault with the authors' choices about this, we do want to point out that the authors' suggestion here that simulation methods cannot mimic data-like clone size distributions or mutation variability (i.e. tree shape) is incorrect. There exist simulation methods that let the user configure these parameters arbitrarily (including using direct inference from data).

We acknowledge that relying on a single dataset to test our method can be risky. Therefore, we use two differently generated datasets, each addressing challenging regions of the parameter space: short CDR3 lengths for our synthetic dataset and high mutation rates for the dataset from [2]. While it is true that other methods also enable repertoire simulation across a broad range of parameters, our primary argument is that our synthetic dataset is more realistic in terms of mutation rate and clone size distributions compared to the dataset used to test the Partis software [2].

[2] Ralph DK, Matsen IV FA (2022) Inference of B cell clonal families using heavy/light chain pairing information. PLOS Computational Biology 18:e1010723.

We have moved the SI figure showing HILARy’s performance across mutation rates on this dataset to the main text (new Figure 5 and Figure 6D-F).

Reviewer 3The authors have addressed the major points I brought up in my review, and the revision is definitely improved. With respect to their new benchmarking on single linkage hierarchical clustering (Figure 4, supplement 2), it's not clear to me why they used their own implementation rather than one of the existing tools, or why the thresholds of 0.76, 0.82, and 0.88 were chosen. However, I think these issues could be addressed with text edits.

We use our implementation as it leverages the prefix tree search method described in the methods section to speed up the single linkage clustering. Given that single linkage clustering relies on a parameter that must be arbitrarily selected based on the repertoire, we conducted a scan for values within the commonly used range. This also highlights the trade-off between precision and sensitivity. Higher thresholds achieve high precision across all mutation rates but sensitivity is degraded for high mutation rates, whereas lower thresholds achieve sensitivity across all mutation rates but precision is degraded for lower mutation rates.

We have added a discussion in the Discussion section.

Personally, I'm trying to reconcile the results of this study with a recent and more comprehensive clonal benchmarking study which showed that single linkage clustering worked quite well: https://bmcimmunol.biomedcentral.com/articles/10.1186/s12865-024-00600-8. May be worth adding some discussion about.

While this study demonstrates that hierarchical clustering yields good results on simulations based on three real datasets, its performance on short CDR3 lengths and high mutation rates remains uncertain, as the mutation loads range from 1.3% to 5.5%.

In Figure S10 A, we illustrate that when using heavy chains only, all hierarchical clustering methods fail to reconstruct clonal families at high precision on short CDR3 lengths in our synthetic dataset and at high sensitivity for high somatic mutation rates in the data from [2]. Furthermore, we have extended the range of CDR3 lengths from 15-45 to 15-60 to show that all methods perform well for large lengths

We have added a discussion of the performance of single-linkage clustering and the above-mentioned paper in the Discussion section.

I agree with Reviewer 2 above points about consistency with AIRR format and that (1) the method seems to perform well in a CDR3 space which represents a small fraction of typical repertoire space, so it's not terribly clear how much it improves overall performance, and (2) it's risky to rely on effectively a single dataset to base these conclusions on.

We address this comment in our reply to reviewer 2 and have added a discussion of these points.